# Prevalence of loss-of-function, gain-of-function and dominant-negative mechanisms across genetic disease phenotypes

Mihaly Badonyi ⓘ ✉ & Joseph A. Marsh ⓘ ✉

Molecular disease mechanisms caused by mutations in protein-coding regions are diverse, but they can be broadly categorised into loss-of-function, gain-of-function and dominant-negative effects. Accurately predicting these mechanisms is important, since therapeutic strategies can exploit these mechanisms. Computational predictors tend to perform less well at the identification of pathogenic gain-of-function and dominant-negative variants. Here, we develop a protein structure-based missense loss-of-function likelihood score that can separate recessive loss of function and dominant loss of function from alternative disease mechanisms. Using missense loss-of-function scores, we estimate the prevalence of molecular mechanisms across 2,837 phenotypes in 1,979 Mendelian disease genes, finding that dominant-negative and gain-of-function mechanisms account for 48% of phenotypes in dominant genes. Applying missense loss-of-function scores to genes with multiple phenotypes reveals widespread intragenic mechanistic heterogeneity, with 43% of dominant and 49% of mixed-inheritance genes harbouring both loss-of-function and non-loss-of-function mechanisms. Furthermore, we show that combining missense loss-of-function scores with phenotype semantic similarity enables the prioritisation of dominant-negative mechanisms in mixed-inheritance genes. Our structure-based approach, accessible via a Google Colab notebook, offers a scalable tool for predicting disease mechanisms and advancing personalised medicine.

The vast majority of disease-causing genetic variants identified to date are located within protein-coding regions of the genome. While many lead to a loss of protein function (LOF), often through premature stop codons or missense changes that destabilise protein folding, others exert their effects via alternative (non-LOF) mechanisms[1]. Gain-of-function (GOF) mutations cause disease through a wide range of molecular mechanisms, including increased activity (hypermorphs), altered binding specificity, or acquisition of novel functions (neomorphs). Dominant-negative (DN) mutations interfere with the activity of the wild-type protein, either by co-assembling into dysfunctional complexes[2] or by competitively sequestering shared binding partners or substrates. Understanding these mechanisms usually requires examining how mutant proteins interact with other molecules. Their impacts can manifest through various means, including disruption or creation of novel interactions, altered binding affinity or specificity, changes to protein complex assembly, and induction of aggregation, mislocalisation, or phase separation[1]. The diversity of molecular mechanisms presents a significant challenge for their identification,

MRC Human Genetics Unit, Institute of Genetics and Cancer, University of Edinburgh, Edinburgh, UK. ✉e-mail: mihaly.badonyi@ed.ac.uk; joseph.marsh@ed.ac.uk

often necessitating elaborate experimental strategies to validate them[3], which are costly and time-consuming.

Accurate prediction and validation of molecular disease mechanisms are essential for developing effective targeted therapies. Diseases resulting from LOF mutations are usually amenable to gene therapy, where the delivery of functional gene copies compensates for the defective allele. This approach has successfully treated conditions such as RPE65-associated retinal dystrophy[4] and Duchenne muscular dystrophy[5]. In contrast, diseases caused by non-LOF mutations are more suited to treatment with small molecules that inhibit the altered or excessive function, as demonstrated by the development of KRAS degraders for cancer[6], or through gene-editing and silencing strategies, as exemplified by a promising treatment for retinitis pigmentosa, driven by the GOF mutation p.Pro23His in rhodopsin[7]. Similar allele-specific targeting approaches offer hope for treating DN conditions, such as collagen-related dystrophy[8] and long QT syndrome[9]. While most genes are associated with a single molecular mechanism, some are known to exhibit multiple mechanisms, requiring distinct therapeutic interventions. For example, sodium channel blockers are effective for epilepsy associated with GOF variants in SCNA1[10], whereas gene replacement therapy may soon address SCNA1 haploinsufficiency in Dravet syndrome[11].

Despite the clear clinical need, predicting molecular disease mechanisms remains difficult. Current computational methods usually focus on predicting LOF and function-altering mechanisms at the level of individual genetic variants[12–14]. However, there are also gene-level features that tend to be associated with different mechanisms[2,15–17]. We recently developed a model to predict the most likely mechanism when heterozygous disease mutations are found in a gene[18]. These predictions have now been incorporated into the DECIPHER database[19], assisting clinicians in identifying potential disease mechanisms.

We previously reported two structural properties—specifically, the energetic impact and clustering of missense variants—that discriminate between genes with LOF and non-LOF mechanisms exceptionally well[16]. This is because LOF mutations tend to be highly destabilising and spread throughout protein structures, whereas non-LOF mutations, which are structurally milder, often exhibit clustering within functionally important regions. We quantify the impacts of variants on protein stability using changes in Gibbs free energy of folding ($\Delta\Delta G$) predicted with FoldX[20], while variant clustering is assessed with the extent of disease clustering (EDC) metric[16]. While $\Delta\Delta G$ is calculated at the variant level, EDC operates at an intermediate level, requiring multiple variants but not necessarily all disease variants within a gene. This flexibility enables EDC to be applied to a group of variants, particularly those associated with the same phenotype, as demonstrated for cancer-associated and Weaver syndrome variants in EZH2[21].

In this study, by integrating EDC and $\Delta\Delta G$ data from pathogenic variants, we develop an empirical distribution-based method to derive a missense LOF (mLOF) likelihood score and demonstrate its utility for improving molecular mechanism predictions in a Bayesian framework. By assembling phenotype annotations for over 70% of pathogenic missense variants in ClinVar[22], we show that the mLOF score is particularly powerful at the phenotype level. Most importantly, we estimate the prevalence of molecular mechanisms across genetic disease phenotypes, revealing widespread mechanistic heterogeneity and highlighting its implications for precision medicine. We make our method available as a Google Colab notebook, allowing mLOF score calculation for variant sets in human protein-coding genes at https://github.com/badonyi/mechanism-prediction.

## Results

### Developing the mLOF score for predicting missense variant molecular mechanisms

Our objective was to predict the likelihood of a set of missense variants being associated with LOF vs. non-LOF molecular mechanisms by integrating information about their protein structural context. Specifically, we sought to combine clustering in three-dimensional space, as quantified by EDC, and predicted energetic impacts, as measured by $\Delta\Delta G$. To achieve this, we developed an approach based on the empirical distributions of these metrics in LOF and non-LOF genes[16], i.e., genes with pathogenic missense variants known to act via LOF and DN or GOF mechanisms, respectively. Importantly, we use $\Delta\Delta G_{rank}$ in place of raw $\Delta\Delta G$ values. This is a recently introduced rank-normalised metric that improves interpretability and facilitates comparisons across different proteins[23,24]. For a given observation of EDC and $\Delta\Delta G_{rank}$ in a set of variants, we calculate the marginal probabilities of these observations being drawn from the LOF rather than non-LOF distributions (Supplementary Fig. 1). The probabilities are then combined into the mLOF score, which represents the likelihood that the variants will have a LOF effect given their energetic impact and dispersal within the protein structure.

To evaluate the utility of the mLOF score, we treated predictions from our previously published proteome-scale model (pDN/GOF/LOF) as informative priors for the likelihood of a disease mechanism occurring in a gene[18]. By updating these priors with the mLOF score, we derived mechanism-specific posterior scores (postDN/GOF/LOF), which represent adjusted estimates of the likelihood that a gene exhibits a mechanism, taking into account the structural properties of its pathogenic missense variants. Figure 1a provides a graphical overview of our method.

We first applied this method to pathogenic missense variants in exclusively autosomal dominant (AD) genes with gene-level molecular mechanism classifications[18], and calculated the area under the receiver operating characteristic curve (AUROC) for the mLOF score, as well as the prior and posterior mechanism-specific scores (Fig. 1b). We found the mLOF score to be predictive across the binary class pairs previously used to construct the priors (DN vs. LOF, GOF vs. LOF, and LOF vs. non-LOF), with AUROC ranging from 0.622 to 0.714, indicating generalisability across the mechanisms.

One possible explanation for the limited performance is that many genes are associated with multiple molecular disease mechanisms, which imposes fundamental limitations on our gene-level approach. Although we only have gene-level rather than phenotype-level classifications, one way of addressing this limitation is by considering those genes with a single disease phenotype, which are thus more likely to be associated with a unique mechanism. Therefore, we used variant-level phenotype annotations from the Online Mendelian Inheritance in Man (OMIM) database[25] to identify dominant genes associated with a single disease phenotype. Notably, AUROC values were markedly increased across all binary class pairs (Fig. 1b). A similar conclusion is supported by the area under the balanced precision-recall curve (AUBPRC) analysis[26] (Supplementary Fig. 2). We also derived the optimal threshold for distinguishing between LOF and non-LOF mechanisms using the single phenotype genes. The resulting value of 0.508 provides a practical cutoff for assessing whether a group of variants is likely to exhibit a LOF mechanism and can be used to compare different variant groups in the same gene. At this threshold, the mLOF score achieves a sensitivity of 0.721, a specificity of 0.702, an accuracy of 0.712, and an F1 measure of 0.719, indicating a balanced performance.

We assessed the robustness of the model in two ways: first, by progressively increasing the minimum number of unique residue positions required for EDC calculation; and second, by restricting the analysis to ClinVar variants with at least a one-star review status. AUROC and AUBPRC values under these conditions are summarised in Table 1. We found that model performance remained stable when limited to variants with at least a one-star review status. As expected, performance moderately improved when more pathogenic residue positions were considered, reflecting increased confidence in the collective properties of the variants.

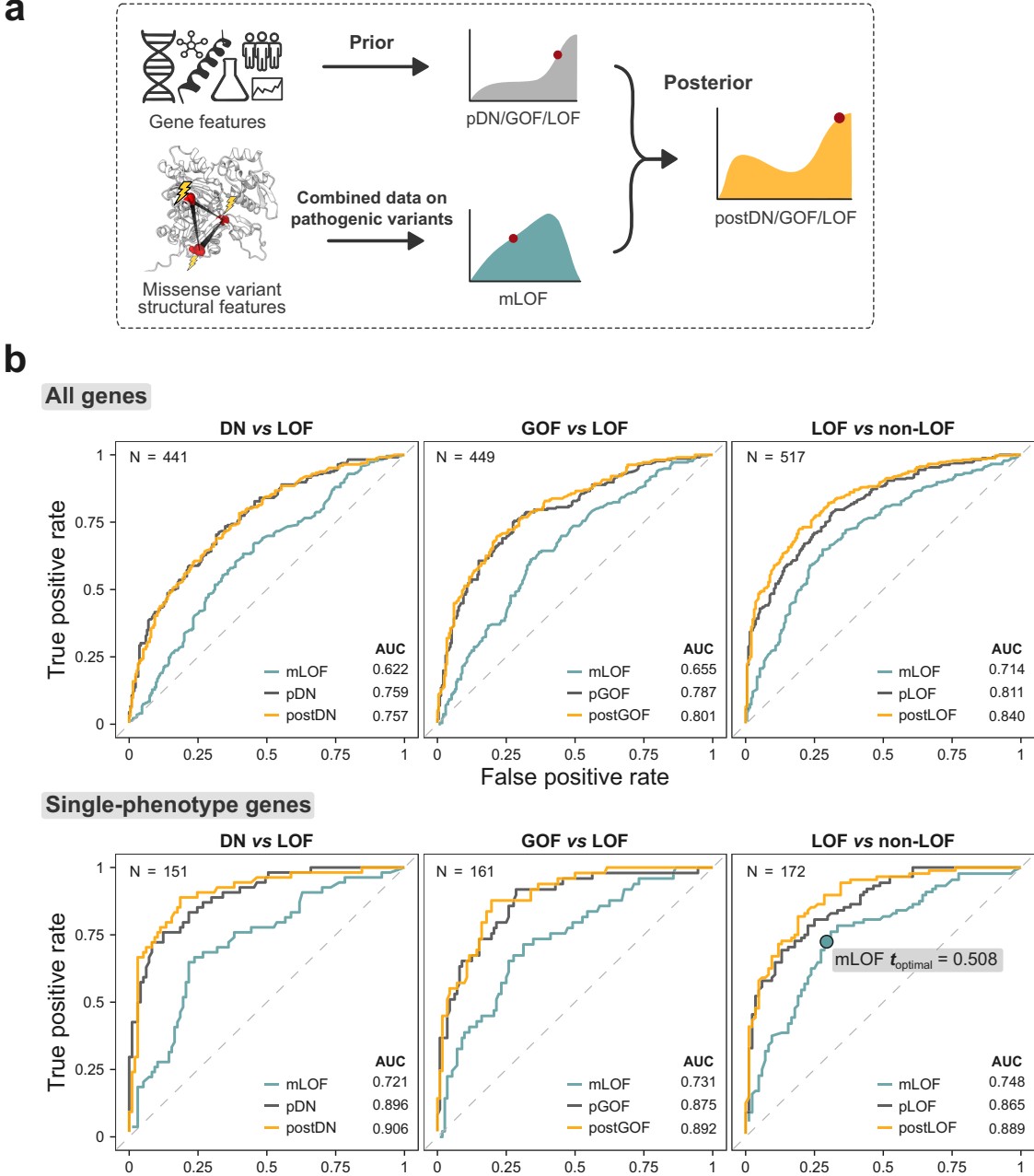

**Fig. 1 | Predicting the likelihood of a loss-of-function mechanism based on the structural properties of pathogenic missense variants. a** Overview of the mLOF score framework. The missense LOF likelihood score mLOF is calculated from empirical distributions of the metrics EDC (spatial clustering) and $\Delta\Delta G_{rank}$ (energetic impact) in LOF and non-LOF genes. This score is then used to update gene-level mechanism-specific priors (pDN/GOF/LOF) established in an earlier study[18]. The final posterior scores (postDN/GOF/LOF) represent adjusted estimates of the likelihood that a gene exhibits a specific molecular disease mechanism, given the structural properties of its pathogenic missense variants. **b** Receiver operating characteristic (ROC) curves and area under the curve (AUC) values of the mLOF score, the prior mechanism probability for the gene, and the posterior mechanism-specific scores across the binary class pairs used to construct the priors. The analysis is split into all genes, using all pathogenic missense variants, and a subset of single-phenotype genes, where only variants linked to the specific OMIM phenotypes are considered. N is the number of genes in each group. mLOF $t_{optimal}$ shows the optimal ROC threshold. Source data are provided as a Source Data file.

As an additional validation, we applied the mLOF score to previously published high-throughput functional assay data on six human disease genes: HRAS[27], MC4R[28], HMBS[29], TP53[30], PTPN11[31], and MTHFR[32] (Supplementary Fig. 3a–f). In all assays, the mLOF score was predictive of the assigned classifications, with scores for the different molecular mechanisms consistently falling above or below the optimal threshold. For example, a clear difference was observed between GOF and LOF HRAS variants, with mLOF scores of 0.426 and 0.613, respectively (Supplementary Fig. 3a). GOF variants were clustered at key functional sites, whereas LOF variants spread across protein core residues. For TP53, we found that variants with a LOF mechanism had the highest mLOF score (0.551; Supplementary Fig. 3d), primarily driven by the dispersal of variants in the structure. DN variants, in contrast, had a lower mLOF score of 0.445 and were concentrated within the DNA-binding domain. Notably, variants exhibiting both DN and LOF properties in the assay clustered exclusively in the DNA-binding domain, showed the highest predicted structural destabilisation, and had the lowest mLOF score (0.351). We speculate that these variants

**Table 1 | Performance of the posterior score postLOF at distinguishing between LOF and non-LOF mechanisms using dominant single-phenotype genes**

| ClinVar P/LP variants | Min N of residues | N genes | AUROC | AUBPRC |
|---|---|---|---|---|
| All, with phenotype | 3 | 172 | 0.889 | 0.874 |
| All, with phenotype | 5 | 112 | 0.927 | 0.910 |
| All, with phenotype | 10 | 47 | 0.937 | 0.911 |
| >= 1-star, with phenotype | 3 | 122 | 0.909 | 0.902 |

The four conditions represent different stringency levels, including the number of unique residue positions (min N of residues) considered for the calculation of the EDC clustering metric or for ClinVar evidence assertion (star-rating). Area under the receiver operating characteristic curve (AUROC) and balanced precision-recall curve (AUBPRC) are shown. Source data are provided as a Source Data file.

are highly destabilising in TP53 knockout assays, but may achieve partial stabilisation through wild-type binding, thus manifesting a DN effect in a context-dependent fashion.

Furthermore, we evaluated the mLOF score against GOF predictions by the LoGoFunc method[12]. Although LoGoFunc provides GOF probabilities at the variant level, averaging these probabilities for a phenotype yields a measure comparable to the mLOF score. We tested the performance of this metric in dominant single-phenotype genes, using both all available genes and the test set of our gene-level predictor. As shown in Supplementary Fig. 3g, h, in both cases, when combined with the prior GOF mechanism likelihood, mLOF yielded postGOF scores that substantially outperformed the average GOF probabilities from LoGoFunc. Notably, although updating pGOF with the average GOF probabilities from LoGoFunc achieved the nominally highest AUROC on all data, its performance declined when evaluated on the test set. We also note that LoGoFunc incorporates many features overlapping with those used to derive the gene-level priors, and is therefore not fully independent of the prior, unlike the mLOF score.

### Prevalence of molecular mechanisms across disease phenotypes

Motivated by these findings, we set out to assess the prevalence of molecular mechanisms across genetic disease phenotypes. We first classified disease phenotypes on the basis of their inheritance. Specifically, genes can show either exclusively autosomal dominant (AD) or autosomal recessive (AR) inheritance, or they may show mixed inheritance, being associated with both dominant and recessive variants. Dominant and recessive variants in mixed-inheritance genes may be associated with distinct phenotypes, in which case we can consider the dominant ([AD]-AD/AR$_{mixed}$) or recessive ([AR]-AD/AR$_{mixed}$) phenotypes separately. In contrast, as we only have gene-level phenotype:inheritance associations available from OMIM, for those genes with mixed-inheritance associated with the same phenotype, we are unable to distinguish between dominant and recessive variants, so they are considered together (AD/AR$_{same}$). We also classified AD phenotypes on the basis of molecular mechanisms, using our previous gene-level LOF, GOF and DN annotations. These phenotype classifications are summarised in Table 2.

Figure 2a shows the mLOF score distribution for the different inheritance-based phenotype classifications, ordered by their mean. The observed distributions align remarkably well with our expectations: AR, [AR]-AD/AR$_{mixed}$ and XLR phenotypes display the highest mLOF scores. AD phenotypes, while shifted to the left side of the optimal threshold by the mean, are evidently bimodal, suggesting the presence of both LOF and non-LOF mechanisms. Interestingly, [AD]-AD/AR$_{mixed}$ phenotypes fall on the left side of the optimal threshold, with a mean of 0.499. This can be explained by considering that the coexistence of a recessive disorder provides a level of evidence against dosage sensitivity[33], thus making AD phenotypes in these genes more likely to arise through alternative mechanisms. In contrast, AD/AR$_{same}$ phenotypes have a higher mean mLOF score relative to AD phenotypes, which could result from the unequal mixing of AD and AR variants—a limitation inherent to our use of phenotype-level rather than variant-level annotations. Alternatively, missense variants in these phenotypes may follow a single

**Table 2 | Inheritance- and mechanism-based classification of disease phenotypes**

| Inheritance-based phenotype classification | Abbreviation |
|---|---|
| Phenotypes of exclusively autosomal recessive genes | AR |
| Autosomal recessive phenotypes of mixed-inheritance genes | [AR]-AD/AR$_{mixed}$ |
| Autosomal dominant phenotypes of mixed-inheritance genes | [AD]-AD/AR$_{mixed}$ |
| Phenotypes of autosomal genes inherited in both dominant and recessive modes | AD/AR$_{same}$ |
| Recessive phenotypes of X-linked genes | XLR |
| Phenotypes of exclusively autosomal dominant genes | AD |
| **Mechanism-based phenotype classification** | |
| Autosomal dominant phenotypes in genes with a loss-of-function mechanism | LOF |
| Autosomal dominant phenotypes in genes with a gain-of-function mechanism | GOF |
| Autosomal dominant phenotypes in genes with a dominant-negative mechanism | DN |
| Autosomal dominant phenotypes in genes without a reported molecular mechanism | Unknown |

inheritance mode, while other mutation types, such as protein null variants (e.g., nonsense or frameshift mutations that are presumed to completely abolish protein function), particularly when homozygous, correspond to the other mode. This phenomenon has been observed, for example, in *ITPR1*, where homozygous null and de novo missense variants both cause Gillespie syndrome[34].

The different mechanism-based phenotype classifications are shown in Fig. 2b. As expected, dominant LOF phenotypes have the highest mean mLOF score (0.547), while GOF and DN phenotypes are strongly left-shifted, with mean mLOF scores of 0.480 and 0.474, respectively. Unknown phenotypes, those of dominant genes without reported mechanisms, show a left-skewed distribution with a mean of 0.484. This likely reflects detection bias, as non-LOF variants are more difficult to experimentally characterise and less well predicted by computational tools, leading to an apparent enrichment of alternative mechanisms in these genes.

Next, we classified AD phenotypes based on their highest mechanism-specific posterior scores into LOF, GOF, and DN categories to assess the contribution of different molecular mechanisms. We focused on three groups in particular: exclusively AD genes with a single phenotype, those with multiple phenotypes, and AD phenotypes in mixed-inheritance genes, i.e., genes associated with both AD and AR disorders. In Fig. 2c, we show the composition of predicted molecular mechanisms across these groups. Single-phenotype AD genes exhibited the largest fraction of phenotypes with a LOF mechanism, at 54.6%. The remaining fraction was attributed to GOF and DN mechanisms occurring at similar frequencies, at 23.8% and 21.6%, respectively. In multi-phenotype AD genes, the fraction of

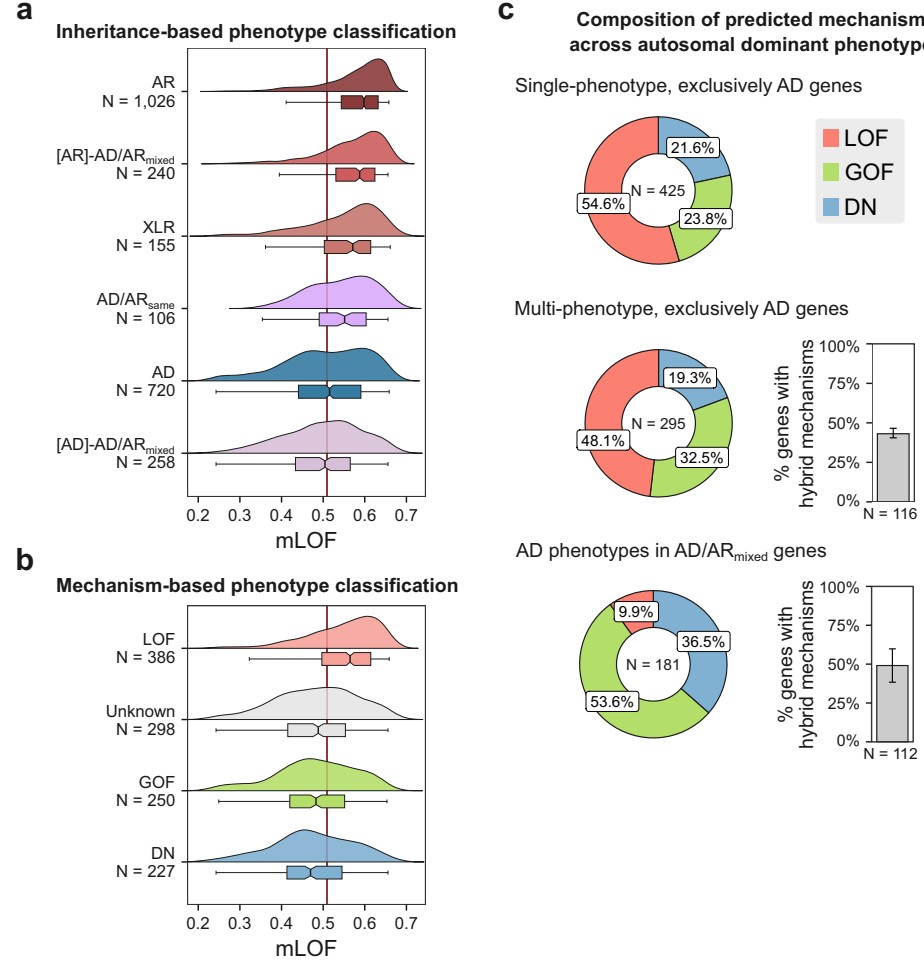

**Fig. 2 | mLOF scores reveal the prevalence of molecular mechanisms at the phenotype level. a, b** Distribution of mLOF scores for inheritance- and mechanism-based phenotype classifications (see Table 2 for the description of the abbreviations). N denotes the number of phenotypes in each group. Red line indicates the optimal mLOF score threshold. Boxes represent data within the 25th and 75th percentiles, the middle line is the median, the notches contain the 95% confidence interval of the median, and the whiskers extend to 1.5× the interquartile range. **c** The fractional composition of predicted mechanisms for phenotypes across the indicated categories. The predictions represent the highest-ranking posterior mechanism-specific score for each phenotype. N denotes the number of phenotypes in each group. Bar charts show the fraction of genes with both LOF and non-LOF ('hybrid') mechanisms, with 50/116 and 55/112 for multi-phenotype AD genes and AD/AR$_{mixed}$ genes, respectively. Error bars are 95% credible intervals calculated from a posterior distribution of fractions derived using the bootstrap estimates of the optimal mLOF threshold. Source data are provided as a Source Data file.

phenotypes with a LOF mechanism was lower, at 48.1%, followed by GOF at 32.5% and DN at 19.3%. This difference may be explained by the observation that multiple disease phenotypes are unlikely to arise in haploinsufficient genes, where reduced dosage (a form of LOF) already causes disease; thus, by exclusion, additional phenotypes are more likely to involve non-LOF mechanisms. AD phenotypes in mixed-inheritance genes had the lowest proportion of LOF mechanisms, at just 9.9%, followed by GOF at 53.6% and DN at 36.5%. As observed with the mLOF score distribution of these genes in Fig. 2a, this likely reflects a reduced likelihood of haploinsufficiency conferred by the presence of a recessive disorder, which makes dominant phenotypes more likely to arise through alternative mechanisms.

We next estimated the fraction of multi-phenotype genes with disease phenotypes involving both LOF and non-LOF molecular mechanisms. This analysis revealed that 43.1% of multi-phenotype AD genes accommodate at least one DN or GOF disease mechanism in addition to LOF. Similarly, in mixed-inheritance genes, we estimated a frequency of 49.1%, assuming most recessive disorders involve biallelic LOF (with rare exceptions[35,36]), and quantifying the fraction with a dominant non-LOF mechanism. These findings suggest that, based on the structural properties of missense variants, mechanistic heterogeneity is widespread among multi-phenotype genes. To facilitate access to these results, we provide a comprehensive list of OMIM phenotypes ($N = 2837$) in Supplementary Data 1, including MIM identifiers, disease names, EDC and $\Delta\Delta G_{rank}$ values, mLOF scores, and the mechanism-specific posterior scores.

**Dominant-negative phenotypes in mixed-inheritance genes**

Intriguingly, our results suggest that LOF is very rare as a mechanism underlying dominant phenotypes in mixed-inheritance genes, accounting for only 9.9% of cases (Fig. 2c). While this might in part be explained by considering that mixed-inheritance genes are less likely to be haploinsufficient, there are many examples where the same phenotype is associated with both dominant and recessive variants. One possible explanation is that the recessive variants are hypomorphic, causing only a partial LOF in each allele that amount to the same net wild-type activity level as complete LOF in one allele. To test this hypothesis, we compared $\Delta\Delta G_{rank}$ distributions of recessive phenotypes in mixed-inheritance genes ([AR]-AD/AR$_{mixed}$) with those in exclusively AR genes (Fig. 3a). We observed that [AR]-AD/AR$_{mixed}$ phenotypes exhibit lower $\Delta\Delta G_{rank}$ values compared with those of AR genes ($P = 1.6 \times 10^{-3}$, Wilcoxon rank-sum test), consistent with the

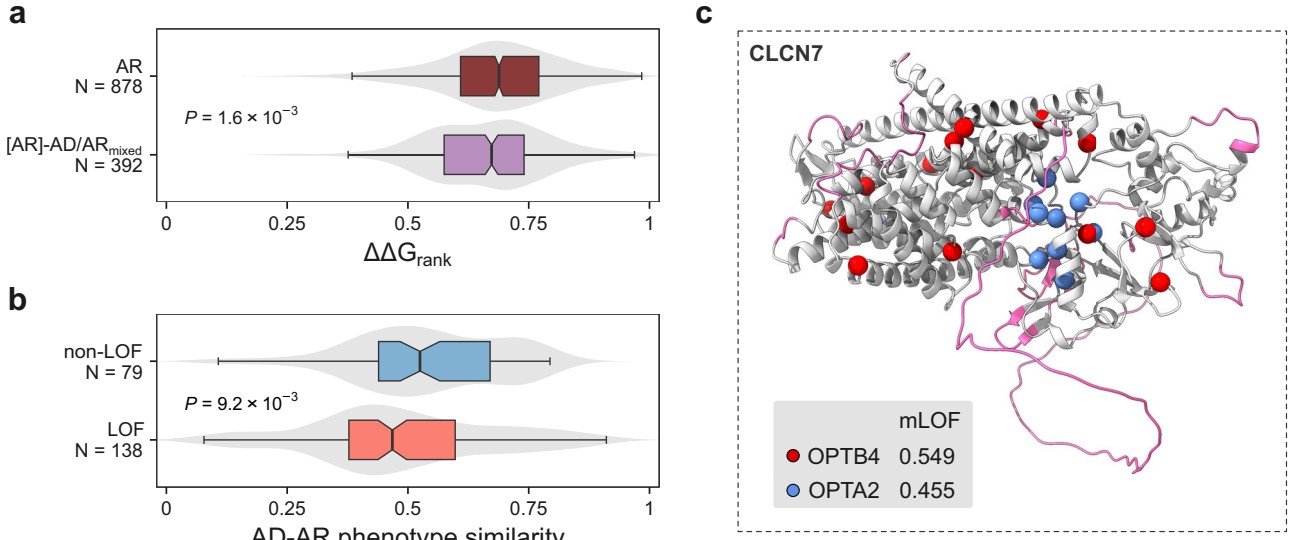

**Fig. 3 | mLOF score and phenotype semantic similarity prioritise dominant-negative phenotypes in AD/AR$_{mixed}$ genes. a** $\Delta\Delta G_{rank}$ distributions for disease phenotypes in exclusively recessive genes (AR) and recessive phenotypes in AD/AR$_{mixed}$ genes ([AR]- AD/AR$_{mixed}$). Boxes represent data within the 25th and 75th percentiles, the middle line is the median, the notches contain the 95% confidence interval of the median, and the whiskers extend to 1.5× the interquartile range. The *P*-value represents a two-sided Wilcoxon rank-sum test. **b** Comparison of AD-AR phenotype semantic similarity within [AR]- AD/AR$_{mixed}$ genes, split into whether the variants of the AD phenotype are predicted to have a non-LOF effect. The *P*-value

represents a two-sided Wilcoxon rank-sum test. **c** *CLCN7*, which encodes the H(+)/ Cl(-) exchange transporter 7, is an example of an AD/AR$_{mixed}$ gene with a reported DN mechanism of pathogenesis. Its structure represents the AlphaFold-predicted model (P51798 [https://alphafold.ebi.ac.uk/files/AF-P51798-F1-model_v1.pdb]). Missense variant positions are shown for the dominant (OPTA2) and recessive (OPTB4) forms of osteopetrosis, with their corresponding mLOF scores displayed below. Regions below a pLDDT of 70 are shown in purple. Source data are provided as a Source Data file.

presence of hypomorphic variants. A similar tendency was observed using raw $\Delta\Delta G$ values (Supplementary Fig. 4). While this trend also appears in AD/AR$_{same}$ genes, we do not have variant-level inheritance classifications in these genes, where the tendency for recessive variants to be hypomorphic may be even stronger, potentially explaining the identical phenotypes for dominant and recessive variants. Nonetheless, the case of *PKD1*, where recessive hypomorphic variants have recently been implicated in polycystic kidney disease[37]—the same phenotype for which there is sufficient evidence of haploinsufficiency caused by dominant LOF mutations (ClinGen[38] Curation ID: 007675)— underscores the relevance of these effects.

Another phenomenon that could explain the tendency for dominant and recessive variants in the same gene to be associated with similar phenotypes is the DN effect, as has been described in cases where DN variants phenocopy recessive disorders[34,39–42]. To test whether AD phenotypes with a predicted non-LOF mechanism have a tendency to phenocopy the recessive disorder, we analysed all non-redundant AD-AR phenotype pairs within AD/AR$_{mixed}$ genes, representing 217 phenotype pairs in 103 genes. These were grouped into a high confidence non-LOF category if the mLOF score for the AD phenotype fell below the optimal threshold and was less than that for the AR phenotype. We then calculated the semantic similarity between AD-AR phenotype pairs using Human Phenotype Ontology[43] terms, with the hypothesis that the non-LOF category should tend to have higher semantic similarity values due to its enrichment in genuine DN mechanisms. As shown in Fig. 3b, the non-LOF class displayed significantly higher AD-AR phenotype similarity values relative to AD-AR phenotype pairs in the LOF class (*P* = 9.2 × 10⁻³, Wilcoxon rank-sum test).

We further refined the analysis by filtering for genes whose DN-specific prior was greater than that for GOF and selecting the phenotype pair with the highest similarity for each gene. Pairs with a semantic similarity greater than 0.5 are listed in Table 3. Among these, we highlight *CLCN7* (Fig. 3c), with an mLOF score of 0.549 for the recessive and 0.455 for the dominant forms of osteopetrosis (OPTB4

and OPTA2, respectively). These scores are reflective of the considerably greater clustering of dominant variants, with an EDC of 1.36 vs. 1.03 for the recessive phenotype. The two phenotypes share many clinical features, as implied by their disease names and semantic similarity scores. Heterozygous osteopetrosis-associated variants are known to exert a dominant-negative effect[44–46]. The p.Gly215Arg variant, for example, disrupts CLCN7 trafficking in a dominant-negative manner[47], and has been used to generate a mouse model of OPTA2, which recapitulates the characteristic osteopetrosis phenotype with excessive bone deposition[48]. These findings highlight the utility of combining mLOF scores with semantic similarity to identify DN disease phenotypes in mixed-inheritance genes.

## Disease phenotypes linked to distinct molecular mechanisms within the same gene

Given that mLOF score analysis suggested considerable mechanistic heterogeneity among multi-phenotype genes, we aimed to identify phenotype pairs most likely to exhibit distinct molecular mechanisms. To this end, we calculated the difference in mLOF scores for all possible phenotype pairs within multi-phenotype genes, excluding those with only recessive inheritance. We further refined our analysis by selecting pairs from beyond the 95th percentile of the distribution, which we consider particularly interesting, and where one phenotype scored above and the other below the optimal threshold. Table 4 summarises these genes, listing their phenotypes with higher mLOF scores (LOF-like) alongside those with lower mLOF scores (non-LOF-like). For some of the top-ranking genes, discussed in more detail below, protein structures and missense variant positions linked to the different phenotypes are shown in Fig. 4.

SMCHD1 (Fig. 4a) is a member of the structural maintenance of chromosomes protein family, which plays an essential role in epigenetic silencing. Mutations in the gene are linked to two distinct clinical phenotypes: the digenic dominant facioscapulohumeral muscular dystrophy type 2 (FSHD2) and the dominant Bosma arhinia

**Table 3 | Top AR-AD phenotype pairs with high semantic similarity, where the dominant phenotype is likely to involve a dominant-negative effect**

| Gene | Recessive phenotype | Dominant phenotype | Similarity |
|---|---|---|---|
| POLG | Progressive external ophthalmoplegia with mitochondrial DNA deletions, autosomal recessive | Progressive external ophthalmoplegia with mitochondrial DNA deletions, autosomal dominant 1 | 0.775 |
| AFG3L2 | Spastic ataxia 5, autosomal recessive | Spinocerebellar ataxia 28 | 0.775 |
| ACTA1 | Congenital myopathy 2B, severe infantile, autosomal recessive | Congenital myopathy 2 C, severe infantile, autosomal dominant | 0.763 |
| ALDH18A1 | Spastic paraplegia 9B, autosomal recessive | Spastic paraplegia 9 A, autosomal dominant | 0.740 |
| TTN | Muscular dystrophy, limb-girdle, autosomal recessive 10 | Myopathy, myofibrillar, 9, with early respiratory failure | 0.708 |
| POLR3B | Leukodystrophy, hypomyelinating, 8, with or without oligodontia and/or hypogonadotropic hypogonadism | Charcot-Marie-Tooth disease, demyelinating, type 1I | 0.700 |
| HTRA1 | Autosomal recessive cerebral arteriopathy with subcortical infarcts and leukoencephalopathy (CARASIL) | Cerebral arteriopathy, autosomal dominant, with subcortical infarcts and leukoencephalopathy, type 2 | 0.669 |
| DEAF1 | Dyskinesia, seizures, and intellectual developmental disorder | Vulto-van Silfout-de Vries syndrome | 0.668 |
| TWNK | Mitochondrial DNA depletion syndrome 7 (hepatocerebral type) | Progressive external ophthalmoplegia with mitochondrial DNA deletions, autosomal dominant 3 | 0.638 |
| CLCN7 | Osteopetrosis, autosomal recessive 4 | Osteopetrosis, autosomal dominant 2 | 0.569 |
| GHR | Laron syndrome | Growth hormone insensitivity, partial | 0.521 |
| LMNA | Mandibuloacral dysplasia | Restrictive dermopathy 2 | 0.511 |
| SAMD9L | Myelodysplasia and leukaemia syndrome with monosomy 7 | Ataxia-pancytopenia syndrome | 0.510 |

Source data are provided as a Source Data file.

microphthalmia syndrome (BAMS). In FSHD2, missense LOF mutations in SMCHD1, combined with a permissive D4Z4 haplotype on chromosome 4, lead to ectopic expression of DUX4, which is toxic to skeletal muscle cells[49]. Conversely, BAMS, characterised by the absence of the nose and accompanied by ocular and reproductive defects, is thought to result from GOF mutations[50]. Structural modelling revealed that BAMS-specific mutations cluster on the protein surface, pinpointing a cryptic interface[51], a finding later confirmed by the crystal structure of the ATPase domain[52]. These observations are borne out by the high mLOF score of BAMS (0.656) and the low mLOF score of FSHD2 (0.274).

KRAS (Fig. 4b) is a signalling protein and established oncogene with GTPase activity. The two phenotypes identified through mLOF analysis are cardiofaciocutaneous syndrome 2 (CFC2) and multiple myeloma. CFC2 is characterised by a distinctive facial appearance, heart defects, and intellectual disability[53]. Heterozygous missense variants underlying the phenotype are dispersed in the protein and have a highly structurally damaging effect, reflected by an mLOF score of 0.623. Supporting this, functional studies on the CFC2-associated variant p.Lys147Glu revealed weak GTP binding, falling short of the oncogenic threshold[54]. In contrast, multiple myeloma variants, which are typically highly recurrent somatic variants[55], cluster around the GTP-binding site and are structurally mild, with an mLOF score of 0.296. Consistent with this, multiple myeloma is strongly linked to KRAS GOF variants[56].

TP63 (Fig. 4c) is a transcription factor required for limb formation from the apical ectodermal ridge[57], linked to two dominant phenotypes: Rapp-Hodgkin syndrome (RHS) and split-hand/foot malformation 4 (SHFM4). RHS is characterised by anhidrotic ectodermal dysplasia and cleft lip and/or palate, and it is associated with LOF mutations in the sterile alpha motif domain (SAM)[58,59]. SHFM4, attributed to GOF mutations[58], presents with clefts in the hands and feet, webbed fingers and toes, underdeveloped bones, and sometimes involves cognitive impairment. In agreement with their reported mechanisms, we found RHS to have a high mLOF score (0.653) due to strongly damaging mutations in the SAM domain, and SHFM4 to have a low mLOF score (0.329) as a result of much milder mutations at solvent-exposed residues. Because TP63 forms tetramers via its oligomerisation domain[60], and may form extended polymeric structures mediated by its SAM domain[61], these structural features could suggest an assembly-mediated GOF (dominant-positive[1]) effect underlying SHFM4. For example, one SHFM4-associated mutation, p.Ala354Glu, is located in a region responsible for interacting with HIPK2[62], which phosphorylates TP63 in response to DNA damage[63].

BRAF (Fig. 4d) is a serine/threonine-protein kinase and an established oncogene in human cancer[64]. Mutations in BRAF are linked to several clinical phenotypes, notably Noonan syndrome 1 (NS1) and multiple myeloma. Missense variants associated with NS1 have an mLOF score of 0.632, suggesting a LOF mechanism. These variants tend to be less clustered but more structurally damaging, and present with cardiac defects, facial dysmorphia, and reduced growth[65]. In contrast, missense mutations linked to multiple myeloma exhibit more activating effects, exemplified by the highly recurrent cancer-driver p.Val600Glu[65,66]. Multiple myeloma variants show a lower mLOF score of 0.321, likely reflective of an underlying GOF mechanism. These variants tend to be milder and localised exclusively within the kinase domain, a region critical for activating downstream signalling in the RAS-MAPK pathway.

MTOR (Fig. 4e) is a serine/threonine protein kinase and the master regulator of cellular metabolism. mLOF score analysis has identified renal carcinoma and CEBALID syndrome (an acronym for craniofacial defects, dysmorphic ears, structural brain abnormalities, expressive language delay, and impaired intellectual development) to have missense variants with dissimilar effects on protein structure. Variants linked to renal carcinoma are dispersed across protein domains and are energetically impactful, yielding an mLOF score of 0.619. In contrast, CEBALID syndrome variants tend to be structurally milder and cluster near the ATP-binding site in the FATC domain, with an mLOF score of 0.32. GOF variants in MTOR have been previously linked to conditions such as Smith-Kingsmore syndrome[67] and there is a growing body of evidence further implicating MTOR in developmental disorders[68-71], with a recent de novo enrichment analysis detecting a significant missense burden in a cohort of in 31,058 parent-offspring trios[72]. Given that two MTOR subunits co-assemble into the mTORC1 complex, these mutations may exert DN or dominant-positive effects, potentially contributing to the observed phenotypic spectrum in MTOR-associated disorders.

AARS1 (Fig. 4f) is the cytoplasmic alanine-tRNA ligase. mLOF score analysis revealed two distinct disease phenotypes: the recessive

**Table 4 | The most different phenotype pairs within multi-phenotype disease genes by the mLOF score**

| Gene | LOF-like phenotype | mLOF | non-LOF-like phenotype | mLOF |
|---|---|---|---|---|
| SMCHD1 | Facioscapulohumeral muscular dystrophy 2 | 0.656 | Bosma Arhinia Microphthalmia Syndrome | 0.274 |
| KRAS | Cardiofaciocutaneous syndrome 2 | 0.623 | Multiple myeloma | 0.296 |
| TP63 | Rapp-Hodgkin syndrome | 0.653 | Split-Hand/foot malformation 4 | 0.329 |
| PRPF8 | Retinitis pigmentosa | 0.656 | Retinitis pigmentosa 13 | 0.339 |
| SMARCB1 | Schwannomatosis | 0.63 | Coffin-Siris syndrome 3 | 0.318 |
| GNAS | Pseudopseudohypoparathyroidism | 0.659 | Pituitary adenoma 3, multiple types, somatic | 0.346 |
| BRAF | Noonan syndrome 1 | 0.632 | Multiple myeloma | 0.321 |
| ABCB11 | Cholestasis, progressive familial intrahepatic 2 | 0.638 | Cholestasis, intrahepatic, of pregnancy 3 | 0.33 |
| CAV3 | Long QT syndrome 9 | 0.598 | Rippling muscle disease | 0.294 |
| MTOR | Renal cell carcinoma, papillary, 1, familial and somatic | 0.619 | CEBALID syndrome | 0.32 |
| ADCY5 | Dyskinesia with orofacial involvement, autosomal recessive | 0.645 | Dyskinesia, familial, with facial myokymia | 0.364 |
| AARS1 | Epileptic encephalopathy, early infantile, 29 | 0.644 | Charcot-Marie-Tooth disease, axonal, type 2 N | 0.366 |
| LRP6 | Tooth agenesis, selective, 7 | 0.656 | Coronary artery disease, autosomal dominant 2 | 0.378 |
| AIFM1 | Deafness, X-linked 5 | 0.585 | Spondyloepimetaphyseal dysplasia, X-linked, with mental deterioration | 0.311 |
| ARX | Mental retardation, X-linked, with or without seizures, ARX-related | 0.637 | Corpus callosum, agenesis of, with abnormal genitalia | 0.368 |
| LHCGR | Hypergonadotropic hypogonadism | 0.567 | Precocious puberty, male | 0.301 |
| TWNK | Perrault syndrome 5 | 0.61 | Progressive external ophthalmoplegia with mito-chondrial DNA deletions, autosomal dominant 3 | 0.346 |
| ACTA1 | Congenital myopathy 2B, severe infantile, autosomal recessive | 0.656 | Congenital myopathy 2 C, severe infantile, auto-somal dominant | 0.395 |
| TP53 | Breast cancer | 0.648 | Medulloblastoma | 0.394 |
| IMPG2 | Macular dystrophy, vitelliform, 5 | 0.648 | Macular dystrophy, vitelliform, 2 | 0.395 |
| HBA2 | Heinz body anemias | 0.651 | Methemoglobinemia, Alpha type | 0.406 |
| SLC32A1 | Generalised epilepsy with febrile seizures plus, type 12 | 0.641 | Developmental and epileptic encephalopathy 114 | 0.397 |
| SCN4A | Congenital myopathy 22 A, classic | 0.62 | Paramyotonia congenita | 0.376 |
| EXT2 | Exostoses, multiple, type II | 0.633 | Ovarian cancer | 0.395 |
| RECQL4 | RAPADILINO syndrome | 0.656 | Ovarian cancer | 0.422 |
| POLR3B | Leukodystrophy, hypomyelinating, 8, with or without oligo-dontia and/or hypogonadotropic hypogonadism | 0.607 | Charcot-Marie-Tooth disease, demyelinating, type 1I | 0.385 |
| APOE | Lipoprotein glomerulopathy | 0.539 | Hyperlipoproteinemia, type III | 0.316 |
| NPR2 | Acromesomelic dysplasia, Maroteaux type | 0.616 | Short stature with nonspecific skeletal abnormalities | 0.395 |
| PRNP | Gerstmann-Straussler disease | 0.552 | Creutzfeldt-Jakob disease | 0.331 |
| FLNA | Cardiac valvular dysplasia, X-linked | 0.652 | Otopalatodigital syndrome, type I | 0.432 |
| TTN | Cardiomyopathy, dilated, 1 G | 0.584 | Myopathy, myofibrillar, 9, with early respiratory failure | 0.365 |
| SDHD | Mitochondrial complex II deficiency, nuclear type 3 | 0.634 | Pheochromocytoma | 0.417 |
| BEST1 | Bestrophinopathy, autosomal recessive | 0.598 | Vitreoretinochoroidopathy | 0.382 |
| MECP2 | Rett syndrome | 0.615 | Mental retardation, X-linked, syndromic 13 | 0.399 |
| TSHR | Hypothyroidism, congenital, nongoitrous, 1 | 0.594 | Ovarian cancer | 0.379 |
| TRPV4 | Metatropic dysplasia | 0.568 | Hereditary motor and sensory neuropathy, type IIC | 0.355 |

Source data are provided as a Source Data file.

developmental and epileptic encephalopathy 29 (DEE29) and the dominant Charcot-Marie-Tooth disease, axonal, type 2 N (CMT2N). Variants associated with DEE29 predominantly map to the ATP-binding site or the acceptor site recognition domain, consistent with its established biallelic LOF mechanism[73]. This is further supported by an mLOF score of 0.644, reflecting the severe structural impact of DEE29-associated mutations. By contrast, CMT2N variants are primarily located in the anticodon-binding domain and in a region homologous to the dimerisation interface observed in a remote paralogue[74]. These variants are associated with a lower mLOF score of 0.366, in agreement with their milder structural effects. Supporting this further, recent studies employing a humanised yeast assay suggest that missense variants linked to CMT2N exert a DN effect[75].

### Mechanism prediction Google Colab notebook

To facilitate mLOF score calculation, we created a Google Colab notebook, available at https://github.com/badonyi/mechanism-prediction, allowing users to input a gene name or UniProt[76] accession number along with a list of variants. The variants should map to the UniProt reference sequence—any mismatch between the variant and the reference amino acid sequence will be flagged with a warning. When only genomic variants are available, we recommend using the ProtVar[76] web server to map these directly to the UniProt canonical isoform. We employ precomputed $\Delta\Delta G_{rank}$ values for the proteome and structures from the AlphaFold database[77] to calculate EDC for the input variants. Although the latter limits proteins to <2700 amino acids, only about 1% of human proteins exceed this length. The results

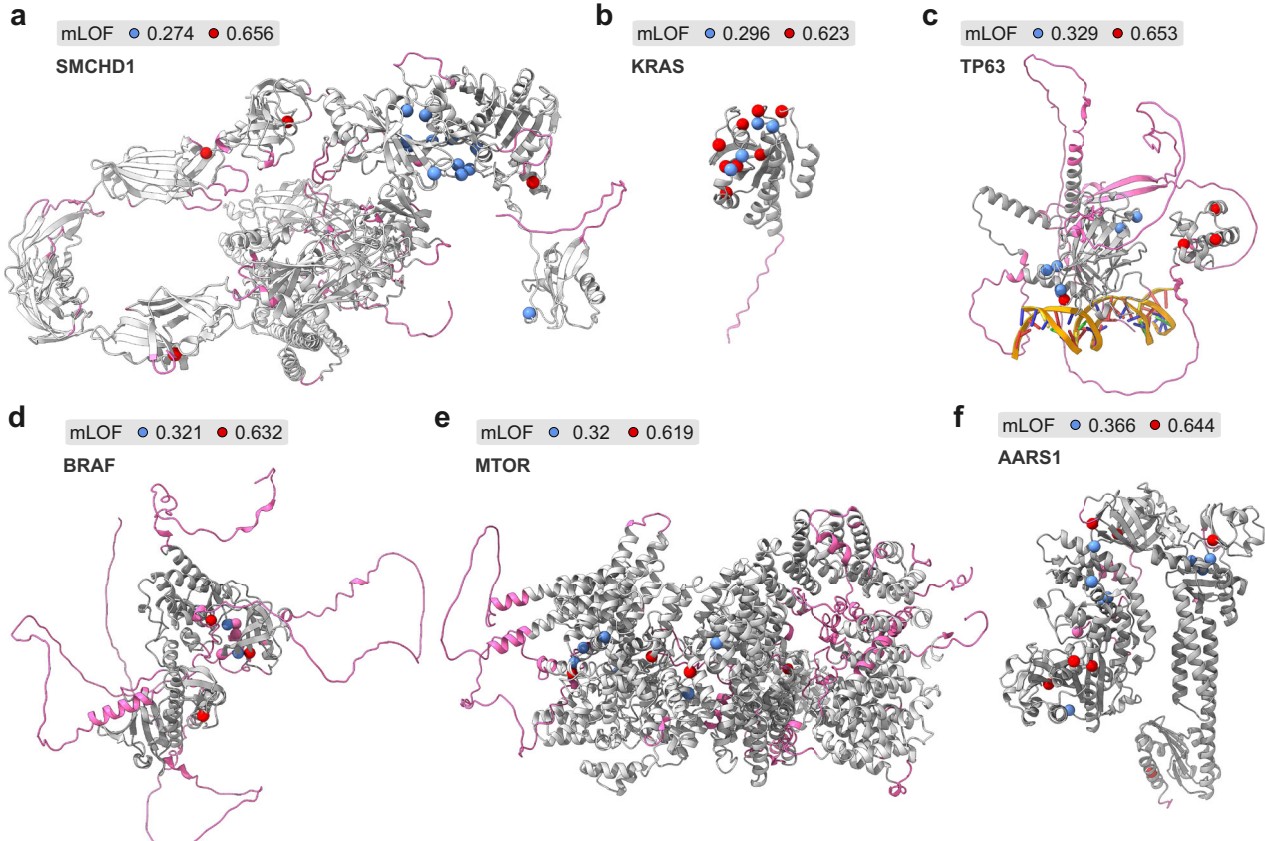

**Fig. 4 | Examples of proteins having two disease phenotypes with mLOF scores indicating both loss-of-function and alternative mechanisms.** Phenotype pairs in top-ranking genes; see main text for a discussion on these and Table 4 for their phenotype definitions. Structures are predicted models from the AlphaFold database. Red and blue spheres represent missense variants associated with the LOF-like and non-LOF-like phenotypes, respectively. Regions below a pLDDT of 70 are shown in purple. **a** AlphaFold model of SMCHD1 (A6NHR9 [https://alphafold.ebi.ac.uk/files/AF-A6NHR9-F1-model_v1.pdb]) **b** AlphaFold model of KRAS (P01116 [https://alphafold.ebi.ac.uk/files/AF-P01116-F1-model_v1.pdb]). **c** AlphaFold model of TP53 (P04637 [https://alphafold.ebi.ac.uk/files/AF-P04637-F1-model_v1.pdb]) superimposed onto the DNA-bound TP63 complex (3USO [https://doi.org/10.2210/pdb3USO/pdb])[122]. **d** AlphaFold model of BRAF (P15056 [https://alphafold.ebi.ac.uk/files/AF-P15056-F1-model_v1.pdb]) **e** AlphaFold model of MTOR (P42345 [https://alphafold.ebi.ac.uk/files/AF-P42345-F1-model_v1.pdb]). **f** AlphaFold model of AARS1 (P49588 [https://alphafold.ebi.ac.uk/files/AF-P49588-F1-model_v1.pdb]).

include all intermediary metrics, such as EDC and $\Delta\Delta G_{rank}$ values, the mLOF and the mechanism-specific posterior scores, as shown in Fig. 5. A brief summary of the results is also provided to assist users in interpreting and reporting their findings.

## Discussion

Here, we developed an empirical distribution-based approach to calculate the missense LOF score, mLOF, which represents the likelihood that a group of pathogenic missense mutations will act via a simple LOF mechanism. We achieved this by leveraging fundamental protein structural properties of missense variants, their energetic impact ($\Delta\Delta G$) and spatial clustering (EDC), both of which have an established and robust relationship to molecular disease mechanisms[16,18,21,24,78]. This approach offers two advantages. First, the use of a non-parametric kernel density estimation method preserves data interpretability at each step, allowing the use of intermediary results for hypothesis testing. Second, the applicability of $\Delta\Delta G$ and EDC to any combination of missense variants provides an optimal metric for assessing the missense LOF likelihood at the phenotype level. Variants within the same gene contributing to the same phenotype are more likely to be causally and functionally coupled, enhancing the precision of molecular mechanism predictions.

In our data, a quarter of genes whose disease phenotypes are linked to missense mutations have more than one associated phenotype. Although this proportion may be skewed by study bias, in that multi-phenotype genes are overrepresented in disease genes that have

historically attracted more attention (e.g., *TP53*, *KRAS*, and *BRCA2*), mLOF score analysis indicates that 43% of dominant and 49% of mixed-inheritance multi-phenotype genes exhibit phenotypes driven by both LOF and non-LOF mechanisms. This finding has important implications for the design of clinical trials and the development of therapeutic interventions, suggesting that in many cases, different disease phenotypes of the same gene may require distinct treatment strategies tailored to the underlying mechanism.

Many dominant phenotypes in mixed-inheritance (AD/AR_mixed) genes are likely attributable to DN effects rather than simple LOF. While this is expected—given that the presence of recessive inheritance reduces the likelihood of haploinsufficiency[33], and a GOF mechanism is unlikely to mimic a recessive disorder[1]—it is nonetheless valuable information from a clinical point of view. By combining mLOF scores with phenotype semantic similarity, we could prioritise phenotypes resembling the recessive disorder in the same gene, identifying cases that may result from DN mechanisms. However, this analysis was not feasible for genes where the same phenotype is inherited in both dominant and recessive patterns (AD/AR_same). In such cases, challenges remain in determining which variants are pathogenic only in the homozygous state and whether dominant variants are as likely to exert DN effects as those in AD/AR_mixed genes.

In many mixed-inheritance genes, the distinction between dominant and recessive modes of action is clear: missense DN variants in *ITPR1* are associated with Gillespie syndrome[34], whereas only recessive null

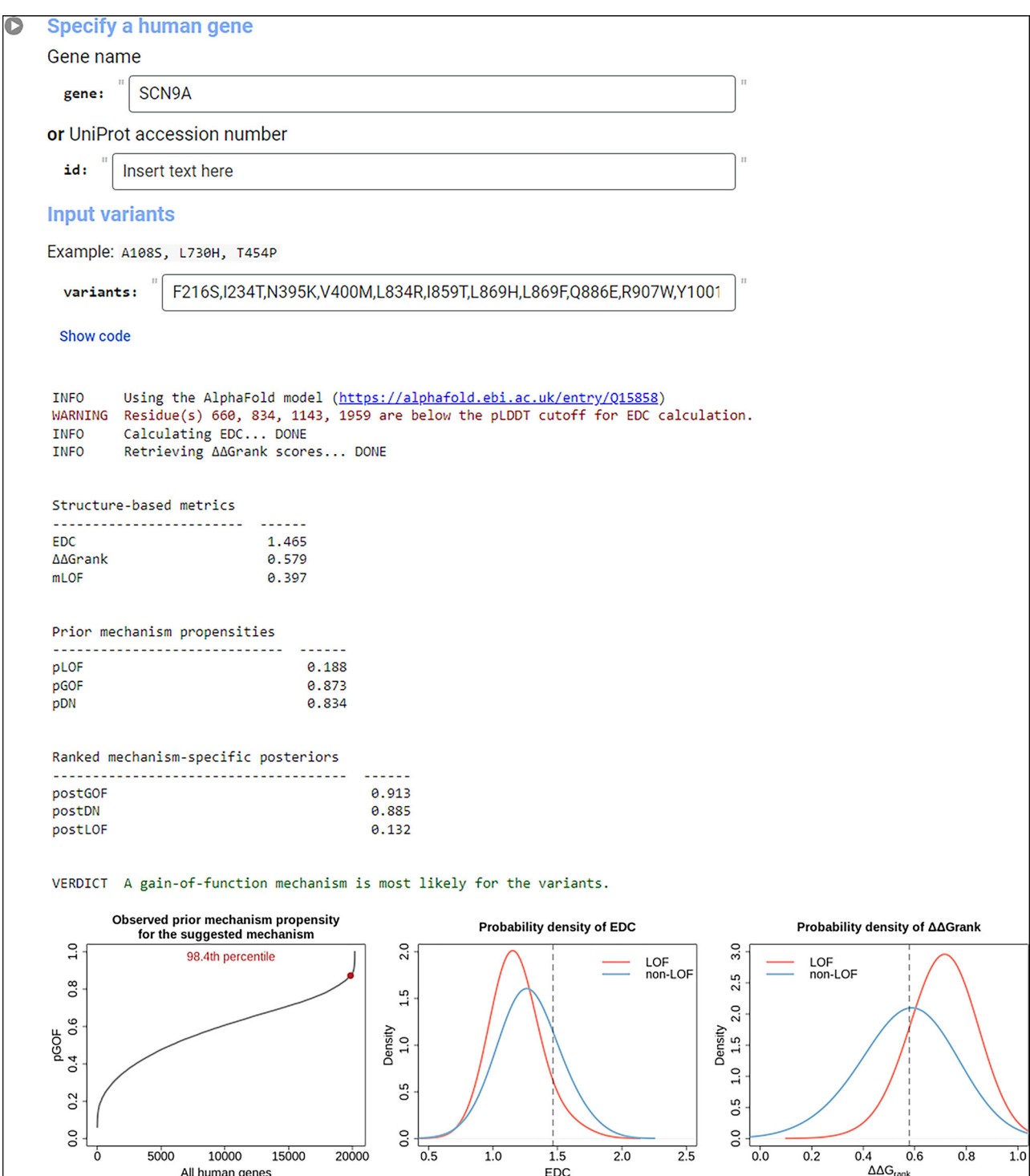

**Fig. 5 | Example input/output of the Colab notebook.** The notebook predicts molecular mechanisms for missense variants by accepting a gene name or UniProt ID along with a list of comma-separated variants in one-letter notation. Results include a link to the gene's AlphaFold model used for EDC calculation, as well as the EDC, $\Delta\Delta G_{rank}$, mLOF score, and prior/posterior mechanism scores. The mechanism with the highest posterior score is suggested. Three plots are generated: (1) where the gene's prior for the suggested mechanism lies relative to all human genes, and (2-3) where the observed EDC and $\Delta\Delta G_{rank}$ values fall within the empirical distributions of LOF and non-LOF genes.

variants have been linked to a clinically identical phenotype[79]. In other cases, however, the distinction is less straightforward. Both recessive and dominant missense mutations in *IGF1R* cause resistance to insulin-like growth factor 1, manifesting in intrauterine growth retardation; for example, the recessive p.Arg40Leu[80] and the dominant p.Val629Glu[81]. Several genetic factors can explain this, e.g., hypomorphic homozygous or compound heterozygous mutations (recessive partial LOF) that

produce phenotypes indistinguishable from those caused by haploinsufficiency (dominant complete LOF). Another factor may be incomplete penetrance[82–85], which causes variable phenotype expressivity within ancestries[83], sexes[86], and even families[87]. For example, the presence of common variants may modify the penetrance of inherited rare variants, lowering the liability threshold required for being affected by a disease[88]. Alternatively, DN variants might present as perfect

phenocopies of the recessive disorder. In some possibly rare cases, a single variant could result in biallelic LOF while exhibiting a DN effect in the heterozygous state, as suggested by emerging evidence in *AARS1*[89].

Considering exclusively dominant genes, diverse genetic and mechanistic factors can explain the coexistence of haploinsufficient and DN phenotypes in the same[90]. For example, allele-specific expression and sex differences in *SMC1A* lead to distinct phenotypes, with truncating mutations linked to haploinsufficiency causing a seizure disorder and DN missense mutations resulting in Cornelia de Lange syndrome[91]. Notably, mutant SMC1A proteins maintain a residual function in males but confer a DN effect in females[92]. A similar phenomenon occurs in *NF1*, where truncating mutations reduce protein levels via haploinsufficiency, while destabilising missense mutations induce a DN effect by promoting degradation of the wild type in a tissue-specific pattern[93]. Although such contrasting mechanisms are scarcer among phenotypes induced only by missense mutations, these examples highlight the nuanced relationship between mutation type, biological context, and the resulting phenotype, complicating the interpretation of molecular mechanisms.

Unlike other molecular mechanism predictors, such as LoGoFunc[12] or VPatho[13], which are typically trained on mechanism labels using supervised learning approaches and may suffer from inflated performance estimates due to circularity issues[94], the mLOF score is a simple, empirical metric independent of existing mechanism classifications. Moreover, the mLOF score also differs in functionality from these predictors. First, it relies on missense variants already known to be causal for a disease or have evidence supporting their involvement in pathogenesis, for example, through family studies or cohort sequencing. Second, rather than inferring the mechanism for a single variant, it harnesses collective properties of variants, which potentially makes its estimate more robust for variants related through their shared phenotype.

Interpreting the posterior mechanism scores requires joint consideration of the prior and mLOF score. In cases where all prior probabilities are uniformly low, the gene may have atypical features not well captured by the training data. Here, a confidently low or high mLOF score may still yield meaningful insight. Conversely, if the prior strongly supports one mechanism but the mLOF score deviates in the opposite direction, this may reflect either limitations in the gene-level model or unusual structural properties of the variant set. We therefore recommend that discordant results be interpreted cautiously and supplemented with orthogonal evidence where possible. This guidance is also included in the Colab notebook, where informative warnings are issued based on score combinations.

Despite the utility of mLOF scores, several limitations remain, many of which could be addressed as more structural data become available. First, our method is necessarily restricted to missense variants, because it relies on EDC and $\Delta\Delta G$, measures that do not readily apply to other mutation consequences like stop-gain, indel, or frameshift variants with respect to differences between molecular mechanisms. Future efforts should focus on developing structure-based methods to evaluate these mutations in a mechanistic context, expanding our variant interpretation ability beyond missense variants. Second, for the EDC calculation, our method includes variants only within regions with a pLDDT (predicted local distance difference test) score[95] above 70, limiting it to non-disordered and well-predicted regions of protein structure. Although pathogenic missense mutations are highly enriched in structured regions[96], this limitation excludes certain variants, e.g., those in short linear motifs[97]. Note that variants in disordered regions can still contribute to mLOF via their predicted $\Delta\Delta G$ impacts. While these values are difficult to interpret quantitatively due to low confidence in AlphaFold models, mutations in disordered regions almost always receive low $\Delta\Delta G$ values. This is consistent with their limited potential to cause global destabilisation and can still be informative for inferring likely molecular mechanisms.

Our approach also assumes that the missense variants used to calculate the mLOF score are causal. This limits its direct application in patient cohorts where often multiple variants in the same gene are identified, and it is difficult to know which variants, if any, can be linked to the disease. In such cases, additional variant prioritisation methods, e.g., the use of variant effect predictors, are required before the mLOF score can be applied.

While structurally mild pathogenic variants often localise to functional sites, such as an interface, they are not exclusively non-LOF. For example, the mutation p.Ile87Arg in PAX6 leads to loss of DNA binding[98,99], causing an aniridia phenotype through haploinsufficiency. Rarely, LOF variants may cluster, as seen in follicular lymphoma-associated mutations in EZH2, which concentrate in the SET domain, disrupting S-adenosyl-L-methionine binding[21]. Conversely, in a few cases, DN variants may be highly structurally damaging. For instance, missense variants linked to late-onset retinal degeneration in C1QTNF5 occur in the C1q head domain, responsible for functional activity, while assembly into a trimeric complex is driven by a separate collagen-like domain. This physical separation may allow co-assembly of wild-type and functionally impaired subunits, despite structural damage in the C1q domain[100].

There are many genes with fewer than three pathogenic missense variants in ClinVar, and when novel disease genes are identified, it is rare for more than a few missense variants to be causally linked to a disease at once. Therefore, our method is primarily applicable to established disease genes whose phenotypes arise through missense variation. Nonetheless, even when the mLOF score cannot be computed, variant location and energetic impact can be informative. For example, low-confidence AlphaFold predictions often coincide with intrinsically disordered regions, where missense mutations are less likely to destabilise the folded structure and, by extension, less likely to act via a canonical LOF mechanism[101]. This may suggest a non-LOF mechanism, particularly when variants occur in close proximity in the sequence or within a well-characterised functional motif. In cases where only one or two variants fall within confidently predicted structured regions, 3D clustering may not be informative, but the energetic impact of individual mutations can still yield mechanistic insight. When interpreted alongside a gene's prior mechanism probabilities, $\Delta\Delta G$ values in structured regions may offer suggestive evidence for a specific mechanism.

Finally, while we currently rely on monomeric AlphaFold models for EDC and $\Delta\Delta G$ estimation, incorporating predicted structures of protein complexes[102–107] could substantially improve the accuracy of missense LOF likelihood estimation by providing more biologically representative structural insights through the consideration of intermolecular interactions[16,108]. For example, our previous work demonstrated that complex properties such as symmetry can be valuable for predicting non-LOF genes, though such features are not yet available for all genes[2]. Improved predictors of protein assembly state and higher-resolution complex structures will likely enhance variant-level predictions. Furthermore, predicting subunits as part of a complex, rather than in isolation, often yields more accurate conformations due to the presence of buried contacts[109], potentially making the spatial clustering of pathogenic residues more sensitive and informative.

In summary, we developed a broadly applicable and readily interpretable metric of missense LOF likelihood, the mLOF score. Our Google Colab notebook offers an accessible platform to compute the score and apply it within a Bayesian framework to predict the most likely mechanism for any combination of pathogenic missense variants in a human gene. This flexibility enables a deeper investigation into the structural effect of mutations, facilitating applications in variant interpretation and molecular mechanism studies.

## Methods

### ClinVar mapping to UniProt reference proteome

Genomic coordinates of 'pathogenic' and 'likely pathogenic' missense variants, which we refer to as pathogenic, were extracted from the ClinVar[22] variant calling file (accessed 10-Sep-2024) using BCFtools[110]. These were subsequently mapped to human reference sequences in

UniProt[76] release 2024_04 with Ensembl VEP 112[111]. We retained phenotype cross-references to the OMIM database, as they represent the most comprehensive and reliable phenotype annotations. These were further annotated with MIM identifiers from two additional sources: the protein-specific JSON files with UniProt variation data via the EBI Proteins API[112], and the UniProt index of human variants curated from literature reports (2024_04 of 24-Jul-2024). Gene-level inheritances were obtained from the OMIM database (06-Aug-2024) via its API. To obtain inheritance modes at the phenotype level, we accessed the phenotype_to_genes.txt and phenotype.hpoa files from the Human Phenotype Ontology (HPO) database[43] (13-Aug-2024), which contain MIM identifiers and their HPO ontology terms. This process resulted in 63,920 pathogenic missense variants, of which 45,888 (71.8%) have an associated OMIM phenotype with an inheritance annotation, as summarised in Supplementary Fig. 5a, b.

## Structural data

We computed EDC and $\Delta\Delta G_{rank}$ based on the predicted human structures downloaded from the AlphaFold database[77,113] (AFDB). For the most part, we used AFDB v1 structures, which are consistent with the 2021_02 UniProt release. Any reference sequence between the lengths of 50 to 5000 amino acids that has undergone a sequence change from the 2021_02 to the 2024_04 UniProt releases were remodelled with AlphaFold2, using the default settings of LocalColabFold (ColabFold[114] v1.5.5) on an NVIDIA A100 GPU with 500 GB of RAM. Structures were visualised using UCSF Chimera X version 1.8[115].

EDC was calculated as previously described[16]. For each residue, we determined the alpha carbon distance to pathogenic residue positions ('disease') and to all other ('non-disease') positions, keeping the shortest distance. The final metric is the ratio of the common logarithm of average non-disease and disease distances. Residues with pLDDT <70 were removed from the calculation, because pathogenic missense mutations are highly enriched in structured regions[96], therefore mutations in disordered proteins with a small structured core may appear clustered relative to the total volume of the model. For proteins modelled as multiple overlapping fragments in the AFDB, we took the fragment with the highest number of missense variants.

To compute $\Delta\Delta G_{rank}$, FoldX 5.0 was first used to estimate the change in the Gibbs free energy for all amino acid substitutions possible by a single nucleotide change based on human codon usage. The RepairPDB command was called on each model before running the BuildModel command to estimate the $\Delta\Delta G$. For pathogenic variants that map to multiple fragments of the same protein in the AFDB, we took the mean $\Delta\Delta G$. The output values were ranked and rescaled so that 0 represents the mildest mutation in the structure and 1 the most damaging. Finally, for any group of variants (e.g., that belonging to a specific phenotype), we average $\Delta\Delta G_{rank}$ values to obtain the mean $\Delta\Delta G_{rank}$ metric, which we refer to as $\Delta\Delta G_{rank}$ for brevity. We note that raw FoldX $\Delta\Delta G$ values are available for non-disordered and well-predicted regions in human AlphaFold models via the ProtVar API[116]. However, as these do not allow calculation of $\Delta\Delta G_{rank}$, we have made our values available at https://osf.io/g98as.

## mLOF calculation

We use genes from Gerasimavicius et al.[16] to fit our model, as these genes have at least one missense variant (rather than, e.g., a protein-truncating variant) associated with a molecular disease mechanism. At both the gene and phenotype levels, we require at least three missense variant positions with a pLDDT >70 to ensure reliable estimates for EDC. We perform Gaussian kernel density estimation separately on the EDC and the $\Delta\Delta G_{rank}$ values of pathogenic missense variants in LOF and non-LOF genes, evaluating at 1024 equidistant points with three times the Sheather-Jones bandwidth[117]. The adjustment factor of three was chosen because, at this value, the probability distributions are smooth and monotonic without noticeable fluctuations, as shown in Supplementary

Fig. 1a, b. To prevent extreme values from disproportionately influencing the probability estimates, we cap the empirical distributions at the 10th and 90th percentiles. We then compute the density functions for both groups and identify the closest point in the density function to each observation, allowing us to derive the estimated density values. $P_{LOF}(EDC)$ and $P_{LOF}(\Delta\Delta G)$, which represent LOF probabilities of observed EDC or $\Delta\Delta G_{rank}$ values, respectively, are computed by dividing the density value of the LOF group by the sum of LOF and non-LOF density values. The combined estimate, i.e., the mLOF score, is obtained by taking the case-specific weighted mean of the two probabilities, which is considered a robust method when the dependence between the variables is strong or unknown[118]. $P_{LOF}(EDC)$ is weighted by the number of variants used for $\Delta\Delta G_{rank}$ calculation, while $P_{LOF}(\Delta\Delta G)$ is weighted by the number of residue positions used for EDC calculation. This approach, which we refer to as the 'weighted mean' method, effectively weakens the influence of $\Delta\Delta G$ when variants are localised to disordered regions, thereby strengthening that of EDC. The previous steps are visually represented in Supplementary Fig. 1. Finally, to estimate a posterior mechanism likelihood score, we use pDN, pGOF, and pLOF from our proteome-scale model as informed priors, which reflect the likelihood of observing the given mechanism when missense variants are identified in a gene[18]. These priors are updated with the mLOF score according to Bayes' rule. We formalise our probabilistic framework below:

*Definitions:*

Let $x$ be a single observation of EDC or mean $\Delta\Delta G_{rank}$.

Let $cap_{LOF}$ and $cap_{non-LOF}$ be the cap values for the observations.

Let $x'$ be the capped observation.

Let $f_{LOF(x')}$ and $f_{non-LOF(x')}$ be the density functions at observation $x'$.

Let $d_{points}$ be the vector of points where the density functions are evaluated.

Let $index$ be the index of the closest value in the density function.

Let $w_{\Delta\Delta G}$ be the number of unique residue positions used for EDC calculation.

Let $w_{EDC}$ be the number of variants used for mean $\Delta\Delta G_{rank}$ calculation.

$$x'(\Delta\Delta G_{rank}) = \begin{cases} cap_{LOF} & \text{if } x > cap_{LOF}, \\ cap_{non-LOF}, & \text{if } x < cap_{non-LOF}, \\ x & \text{otherwise,} \end{cases} \quad (1)$$

$$x'(EDC) = \begin{cases} cap_{LOF} & \text{if } x < cap_{LOF}, \\ cap_{non-LOF} & \text{if } x > cap_{non-LOF}, \\ x & \text{otherwise,} \end{cases}$$

*Finding indices of nearest density points for a given capped observation $x'$:*
$$index_{LOF}(x') = \arg\min_i |d_{points}[i] - x'|$$
$$index_{non-LOF}(x') = \arg\min_i |d_{points}[i] - x'|$$
$$(2)$$

*Obtaining density values:*
$$f_{LOF}(x') = f_{LOF}\left(d_{points}\left[index_{LOF}(x')\right]\right) \quad (3)$$
$$f_{non-LOF}(x') = f_{non-LOF}\left(d_{points}\left[index_{non-LOF}(x')\right]\right)$$

*Calculating $P_{LOF}(EDC)$ and $P_{LOF}(\Delta\Delta G)$:*
$$P_{LOF}(x') = \frac{f_{LOF}(x')}{f_{LOF}(x') + f_{non-LOF}(x')} \quad (4)$$

*Calculating the mLOF score:*
$$mLOF = \frac{w_{EDC} \cdot P_{LOF}(EDC) + w_{\Delta\Delta G} \cdot P_{LOF}(\Delta\Delta G)}{w_{EDC} + w_{\Delta\Delta G}} \quad (5)$$

*Calculating mechanism−specific posterior scores*:

$$postDN = \frac{(1 - mLOF) \cdot P_{DN}}{(1 - mLOF) \cdot P_{DN} + mLOF \cdot (1 - P_{DN})}$$

$$postGOF = \frac{(1 - mLOF) \cdot P_{GOF}}{(1 - mLOF) \cdot P_{DN} + mLOF \cdot (1 - P_{GOF})} \quad (6)$$

$$postLOF = \frac{mLOF \cdot P_{LOF}}{mLOF \cdot P_{LOF} + (1 - mLOF) \cdot P_{LOF}}$$

## Method validation

We initially compared the performance of the weighted mean method to a generalised linear model (GLM) that estimates mLOF from EDC and $\Delta\Delta G_{rank}$ and an interaction term between them. The rationale was that a GLM may better capture the joint distribution of the metrics, potentially outperforming the weighted mean method, which relies on marginal distributions. By comparing bootstrapped AUROC estimates, we found that the posterior mechanism-specific scores obtained with the GLM-based mLOF score had a consistently worse performance across the binary class pairs. A possible explanation for this is that mLOF scores from the GLM (Supplementary Fig. 1d) are much less conservative than those from the weighted mean model (Supplementary Fig. 1c), leading to the mLOF score having a greater influence on the posterior. This result suggested that a generalised linear model cannot achieve the same performance as our weighted mean model, supporting its use.

To evaluate the mLOF score's utility in distinguishing different molecular mechanisms within the same gene, we applied it to missense LOF, GOF, DN, and hyper-complementing (HyC) variants detected by multiplexed assays of variant effect (MAVEs). We identified six MAVEs in which at least two of the aforementioned molecular disease mechanisms, or functional consequences in the case of HyC variants, have been confidently detected in the same gene: *TP53*[30], *HRAS*[27], *MC4R*[28], *HMBS*[29], *TP53*[30], *PTPN11*[31], and *MTHFR*[32]. All scores were obtained from the supplementary material of the respective publication.

For TP53, we adopted the classification approach described in the original study: DN and LOF variants were defined as those with Z-scores three standard deviations (SD) from the mean of all synonymous mutations, based on the 'p53WT+nutlin-3' assay for DN variants and the 'p53NULL+nutlin-3' and 'p53NULL+etoposide' assays for LOF variants. For visualisation in Supplementary Fig. 3d, a combined score was calculated as ('p53WT+nutlin-3' + 'p53NULL+nutlin-3' – 'p53NULL+etoposide')/3.

For HRAS, we defined LOF variants as those with relative enrichment values more than two SD below the mean in the 'DMS_regulated' assay, and GOF variants as those more than two SD above the mean in the 'DMS_attenuated' assay. We note that this classification is more stringent than the one SD threshold used by the authors. For visualisation in Supplementary Fig. 3a, the combined score was calculated as ('DMS_regulated' + 'DMS_attenuated' + 'DMS_unregulated')/3.

For MC4R, we selected the 'THIQ 1.2e-08_minus_Forsk_2.5e-05' contrast for analysis, as GOF variants are most discernible under low agonist stimulation. LOF and GOF variants were defined as those with 'log2ContrastEstimate' values more than two SD below and above the mean, respectively.

For HMBS ('score' column) and MTHFR ('base functionality' column), missense LOF and HyC variants were classified relative to the score distribution of synonymous variants. LOF variants were defined as those scoring below the mean minus two SD, and HyC variants as those exceeding the mean plus two SD.

For PTPN11, we used the 'Enrichment (ave)' column and defined LOF and GOF variants as those more than two SD below and above the mean of the distribution for missense variants, respectively.

For the analysis with LoGoFunc-predicted GOF and LOF variants[12], we downloaded genome-wide missense variant predictions from the

GOF/LOF database (release 10-Aug-2024, https://itanlab.shinyapps.io/goflof/). Genomic coordinates of these variants were mapped using the Ensembl VEP 112 pipeline (as described above) and merged with our ClinVar dataset. We computed the mean LoGoFunc_GOF score for variants associated with phenotypes in single-phenotype AD genes and evaluated it against all genes in our GOF vs. LOF dataset, as well as the corresponding test set. The test set excludes genes used for training the model (used to construct pGOF) and is limited to proteins with <50% pairwise sequence identity.

## Phenotype-level analyses

To ensure reliable mLOF estimates, we considered disease phenotypes with missense variants at 3 distinct positions. In Fig. 2a, b, the following criteria were used to create the phenotype classifications. Note that gene-level molecular mechanisms are based on our previous study[18].

1. **AR**: the gene is exclusively AR in OMIM, and the phenotype is annotated as AR in HPO.
2. **[AR]-AD/AR_mixed**: the gene has at least one AD and one AR phenotype with a sufficient number of missense variants, and the phenotype is annotated as AR in HPO.
3. **[AD]-AD/AR_mixed**: the gene has at least one AD and one AR phenotype with a sufficient number of missense variants, and the phenotype is annotated as AD in HPO.
4. **AD/AR_same**: the phenotype has both AD and AR inheritance annotation in HPO.
5. **XLR**: any phenotype of genes in OMIM with 'X-linked recessive' or 'X-linked' inheritance.
6. **AD**: the gene is exclusively AD in OMIM, and the phenotype is annotated as AD in HPO.
7. **LOF**: the phenotype is annotated as AD in HPO, with the gene either having a reported LOF mechanism or has 'Sufficient evidence for dosage pathogenicity' in the ClinGen database[38] as of 10-Sep-2024. Does not overlap with DN or GOF genes.
8. **GOF**: the phenotype is annotated as AD in HPO, and the gene has a reported GOF disease mechanism. Excludes AD/AR_mixed genes. May overlap with DN genes.
9. **DN**: the phenotype is annotated as AD in HPO and the gene has a reported GOF disease mechanism. Excludes AD/AR_mixed genes. May overlap with GOF genes.
10. **Unknown**: the gene is exclusively AD in OMIM, lacks a reported disease mechanism, and the phenotype is annotated as AD in HPO.

For each within-gene phenotype pair, we calculated how missense variant sets relate to each other in terms of overlap: (1) distinct, if the variant sets are mutually exclusive; (2) intersect, where some variants are shared between the sets; (3) subset, if variants of one phenotype represent a subset of the other; and (4) identical, if the variant sets are mutually inclusive. Supplementary Fig. 5c illustrates the relative proportions of set relationships across all non-redundant phenotype pairs and within unique inheritance groupings. As the mLOF score can be affected by the extent of variant overlap, losing discriminatory value for 'identical' sets, we only considered phenotype pairs whose variants had 'distinct' and 'intersect' set relationships.

Semantic similarity between AD-AR phenotype pairs was calculated with the ontologyIndex and ontologySimilarity R packages based on the 08-Feb-2024 HPO release, using Lin's expression of term similarity[119].

## Statistical analysis

Data analysis was performed in R 4.3.0[120], using the tidyverse meta-package. Statistical tests were two-sided, and an alpha level of 0.05 was considered significant. In bootstrap analyses, 1,000 resamples were used. The optimal threshold was derived by selecting the value that minimises the combined Euclidean distance from the (0,1) coordinate of

the ROC curve, based on the true positive and false positive rates. Balanced precision was computed by adjusting the standard precision to account for class imbalance, following a previously introduced formula[26].

**Reporting summary**

Further information on research design is available in the Nature Portfolio Reporting Summary linked to this article.

## Data availability

All raw datasets used in this study can be accessed from the OSF repository (DOI: 10.17605/OSF.IO/AH2UC)[121], available at https://osf.io/ah2uc. A README file describing each dataset is available at https://osf.io/5w3qf. AlphaFold-predicted structures, including those shown in Fig. 3c and Fig. 4, can be accessed from the AlphaFold Protein Structure Database at https://ftp.ebi.ac.uk/pub/databases/alphafold/v1/UP000005640_9606_HUMAN_v1.tar. Those predicted structures that underwent a sequence change between the 2021_02 and 2024_04 UniProt releases and were generated in this study are available in PDB format at https://osf.io/e32q9. $\Delta\Delta G_{rank}$ values for all missense variants in the human proteome, based on AlphaFold-predicted structures, can be downloaded in bulk at https://osf.io/g98as. Source data are provided with this paper. Previously published databases or datasets used in this work: ClinVar (accessed 10-Sep-2024) (https://www.ncbi.nlm.nih.gov/clinvar/), dataset: https://osf.io/9e3h2; ClinGen haploinsufficiency curations (https://clinicalgenome.org/), dataset: https://osf.io/2cze7; EBI Proteins API; UniProt humsavar (2024_04 of 24 Jul 2024) https://ftp.uniprot.org/pub/databases/uniprot/current_release/knowledgebase/variants/humsavar.txt); Online Mendelian Inheritance in Man database (06-Aug-2024) (https://www.omim.org/api/), dataset: https://osf.io/cdnqy; Human Phenotype Ontology database (13-Aug-2024) (https://hpo.jax.org/), datasets: https://osf.io/9e2pn and https://osf.io/46yxv; AlphaFold Protein Structure Database (https://alphafold.ebi.ac.uk/), dataset: https://ftp.ebi.ac.uk/pub/databases/alphafold/v1/UP000005640_9606_HUMAN_v1.tar; LoGoFunc predictions (https://itanlab.shinyapps.io/goflof/), dataset: https://osf.io/wmesg; TP53 deep mutational scanning data (https://doi.org/10.1038/s41588-018-0204-y), dataset: https://osf.io/wntsv; HRAS deep mutational scanning data (https://doi.org/10.7554/eLife.27810), dataset: https://osf.io/5jw9k; MC4R deep mutational scanning data (https://doi.org/10.7554/elife.104725), dataset: https://osf.io/b2r9q; HMBS deep mutational scanning data (https://doi.org/10.1016/j.ajhg.2023.08.012); dataset: https://osf.io/e3fgm; PTPN11 deep mutational scanning data (https://doi.org/10.1101/2024.05.13.593907); dataset: https://osf.io/32hfg; MTHFR deep mutational scanning data (https://doi.org/10.1016/j.ajhg.2021.05.009); dataset: https://osf.io/4e2jz; Crystal structure of TP53: 3USO AlphaFold model of SMCHD1 (A6NHR9) AlphaFold model of KRAS (P01116) AlphaFold model of TP53 (P04637) AlphaFold model of BRAF (P15056) AlphaFold model of MTOR (P42345) AlphaFold model of AARS1 (P49588) Source data are provided with this paper.

## Code availability

Code to reproduce all analyses is available at https://osf.io/ah2uc. The Colab notebook for mechanism prediction can be accessed at https://github.com/badonyi/mechanism-prediction. A copy of this notebook has also been deposited in the repository associated with this project and can be found at https://osf.io/27wc4.

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

## Acknowledgements

We thank Ben Livesey for his helpful comments on the manuscript. This project was supported by funding from the European Research Council (ERC) under the European Union's Horizon 2020 research and innovation programme (grant agreement No. 101001169) and by funding from the Medical Research Council (MRC) Human Genetics Unit core grant (MC_UU_00035/9). JAM is a Lister Institute Research Prize Fellow. This work has made use of the resources provided by the Edinburgh Compute and Data Facility (ECDF).

## Author contributions

M.B. performed the analyses under the supervision of J.A.M. M.B. and J.A.M. wrote the manuscript.

## Competing interests

The authors declare no competing interests.
