## [Transparent Peer Review file · Nature Communications]

Prevalence of loss-of-function, gain-of-function and dominant-negative mechanisms across genetic disease phenotypes

Corresponding Author: Professor Joseph Marsh

Version 0:

Reviewer comments:

Reviewer #1

(Remarks to the Author)

The manuscript "Prevalence of loss-of-function, gain-of-function and dominant negative mechanisms across genetic disease phenotypes" details an empirically derived metric indicating the likelihood missense variants contribute to disease via LOF or non-LOF mechanism. This metric is informed by structural properties, change in free energy ($\Delta\Delta G$) and a measure of variant clustering (EDC), that have established relationship with disease mechanism. Scores from this model are used to update gene-level priors indicating a genes association with a given molecular mechanism (LOF, GOF, DN). The posterior scores exhibit modest improvement for the identification of molecular mechanisms.

The author's metric mLOF is valuable in its applicability to sets of related variants, e.g. those resulting in a given phenotype within a gene. This may add information beyond variant-level classification in some cases. Further, via combination with their gene-level classifier, the authors note intriguing and important observations regarding the prevalence of mechanisms in genes with different inheritance patterns. Some questions/comments:

- 1) The mLOF score is reliant on at least three suspected variants resulting in a shared phenotype and occurring in non-disordered, well-predicted regions of AlphaFold models. In practice, especially for novel disease genes or gene-phenotype pairs, this limitation may preclude use. The authors should discuss how results should be interpreted in such cases and how users might maximize utility for novel analyses.
- 2) The authors should discuss interpretation of the posterior scores when the priors are low-confidence and/or the mLOF adjustment contradicts the prior.
- 3) The metric is heavily dependent on high-quality pathogenicity and phenotype attributions. Yet the authors do not indicate that low-confidence ClinVar assertions are filtered or otherwise accounted for. This must be addressed and/or discussed.
- 4) The functional validation is interesting. Why is this not expanded to other DMS assaying stability in ProteinGym?

Additional considerations

1) The authors demonstrate that the mLOF update to the pGOF prior (mLOF + pGOF) results in better performance. In Figure S3D, it is also shown that LoGoFunc outperforms mLOF on its own. Why not assess LoGoFunc for updating the priors (LoGoFunc + pGOF). Does postGOF improve, remain the same, deteriorate?

(Remarks on code availability)

Reviewer #2

(Remarks to the Author)

Badonyi et al. is building on the findings they described in their previous paper that shows that computational predictors tend to perform poorly on identifying GOF or DN pathogenic variants. Here, they develop a predictive score (mLOF) to distinguish LOF mechanisms from other disease mechanisms. It is an empirical distribution-based approach, which computes the likelihood that a group of missense variants acts via LOF mechanism. The method takes a priori identified pathogenic variants as an input and does not explicitly classify pathogenicity of variants but the mechanism. The mLOF scores are based on their previously described clustering of pathogenic variants (EDC) based on AlphaFold structures and estimated stability effects of mutations using FoldX and computing a normalized mutation effect (ddGrank). The performance of their score is reasonable both on curated clinical data and functional assays of TP53 and HRAS proteins. And it performs better than LoGoFunc method. They share it as a Google Colab notebook that is easy to use, and will be a valuable resource for the biomedical and geneticist community.

Using mLOF scores they find that half of the phenotypes in dominant genes are actually DN or GOF, that is a high estimate. I find the paper well written, easy to follow with clear figures and representations. The work is novel and interesting and highlights the mechanistic heterogeneity of disease mechanisms even within the same gene posing a challenge for variant interpretation and personalized medicine.

Specific comments:

I would include the definition of gain-of-function and dominant negative effects in the introduction.

I wonder if a cutoff of 3 variants only is not too low to compute EDC? Please show that the results are robust to the choice of this threshold.

Regarding performance, please report different metrics at the relevant threshold of 0.508, e.g. sensitivity, specificity, F1 score, accuracy beyond AUROC and precision-recall curve.

Line 166: what is a homozygous null variant, please define. This statement does not support the previous claim regarding other mutation types than missense. In general "null variants" and "LOF variants" terms are both used throughout the text. If they are treated as synonyms maybe using only LOF would make the text cleaner.

Line 185: it is unclear how haploinsufficiency is related to non-LOF mechanisms.

The scores for POLG do not seem to be that different or recessive and dominant forms (0.518 and 0.446) and also visually in Figure 3C there is not a big difference in clustering of variants.

It would be interesting to see the distribution of ddG values and not only the ranked (normalized) score as well for the different gene types stratified by inheritance modes (like in Figure 3AB).

Since disordered regions (defined by pLDDT <70) and the variants that fall within were excluded from all the analyses, indicating those regions in the structural representation would be helpful, eg. In Figure 4.

Please provide the sample size in the caption or plot for Figure S2 and refer to the relevant dataset used. How was balanced precision achieved?

In Figure S3C and D it is not clear which dataset was used. Please reference it.

In Figure S3A, it is unclear what the mLOF numbers in the legend refer to. Are these the mean values? Maybe showing the distribution would be more meaningful?

I do not understand what is shown in Figure S4C and D and I did not find a reference to these panels in the text.

I wonder if incorporating structural data on protein complexes would improve the annotation, especially for dominant negative variants that would likely interfere with the assembly of protein complexes for instance. It may be outside of the scope of this work though.

(Remarks on code availability)

I tested the Google Colab notebook and it worked fine.

Version 1:

Reviewer comments:

Reviewer #1

(Remarks to the Author)

The authors have appropriately addressed all my concerns.

(Remarks on code availability)

Reviewer #2

(Remarks to the Author)

The authors have answered all my questions and clarified them in the text. I have no further concerns, only a comment about disordered regions.

Originally, I thought that disordered regions (< pLDDT score 70) were excluded from the analysis. But in the response letter the author stressed that they are only excluded from the EDC analysis.

The authors say (line 438)

“disordered regions can still contribute to mLOF via their predicted $\Delta\Delta G$ impacts”

I am questioning how reliable these $\Delta\Delta G$ values are in these flexible regions that are predicted with low confidence by AlphaFold. Please add some benchmarking or cite previous work that validates the use of FoldX in intrinsically disordered/low pLDDT regions. Or warn the users that these values have to be interpreted with caution.

(Remarks on code availability)

Response to Reviewers

Reviewer #1 (Remarks to the Author):

The manuscript “Prevalence of loss-of-function, gain-of-function and dominant negative mechanisms across genetic disease phenotypes” details an empirically derived metric indicating the likelihood missense variants contribute to disease via LOF or non-LOF mechanism. This metric is informed by structural properties, change in free energy ($\Delta\Delta G$) and a measure of variant clustering (EDC), that have established relationship with disease mechanism. Scores from this model are used to update gene-level priors indicating a genes association with a given molecular mechanism (LOF, GOF, DN). The posterior scores exhibit modest improvement for the identification of molecular mechanisms.

The author’s metric mLOF is valuable in its applicability to sets of related variants, e.g. those resulting in a given phenotype within a gene. This may add information beyond variant-level classification in some cases. Further, via combination with their gene-level classifier, the authors note intriguing and important observations regarding the prevalence of mechanisms in genes with different inheritance patterns. Some questions/comments:

1) The mLOF score is reliant on at least three suspected variants resulting in a shared phenotype and occurring in non-disordered, well-predicted regions of AlphaFold models. In practice, especially for novel disease genes or gene-phenotype pairs, this limitation may preclude use. The authors should discuss how results should be interpreted in such cases and how users might maximize utility for novel analyses.

We have now added a new paragraph discussing this limitation in the **Discussion**:

“There are many genes with fewer than three pathogenic missense variants in ClinVar and, when novel disease genes are identified, it is rare for more than a few missense variants to be causally linked to a disease at once. Therefore, our method is primarily applicable to established disease genes whose phenotypes arise through missense variation. Nonetheless, even when the mLOF score cannot be computed, variant location and energetic impact can be informative. For example, low-confidence AlphaFold predictions often coincide with intrinsically disordered regions, where missense mutations are less likely to destabilise the folded structure and, by extension, less likely to act via a canonical LOF mechanism. This may suggest a non-LOF mechanism, particularly when variants occur in close proximity in the sequence or within a well-characterised functional motif. In cases where only one or two variants fall within confidently predicted structured regions, 3D clustering may not be informative, but the energetic impact of individual mutations can still yield mechanistic insight. When interpreted alongside a gene’s prior mechanism probabilities, $\Delta\Delta G$ values in structured regions may offer suggestive evidence for a specific mechanism”.

2) The authors should discuss interpretation of the posterior scores when the priors are low-confidence and/or the mLOF adjustment contradicts the prior.

We have now added the following paragraph to the **Discussion**:

“Interpreting the posterior mechanism scores requires joint consideration of the prior and mLOF score. In cases where all prior probabilities are uniformly low, the gene may have atypical features not well captured by the training data. Here, a confidently low or high mLOF score may still yield meaningful insight. Conversely, if the prior strongly supports one mechanism but the mLOF score deviates in the opposite direction, this may reflect either limitations in the gene-level model or unusual structural properties of the variant set. We therefore recommend that discordant results be interpreted cautiously and supplemented with orthogonal evidence where possible. This guidance is also included in the Colab notebook, where informative warnings are issued based on score combinations”.

3) The metric is heavily dependent on high-quality pathogenicity and phenotype attributions. Yet the authors do not indicate that low-confidence ClinVar assertions are filtered or otherwise accounted for. This must be addressed and/or discussed.

We agree that the quality of pathogenicity and phenotype annotations is a critical consideration. We have clarified this in the revised manuscript and, in response to another reviewer comment, combined it with our assessment of the robustness of EDC to the choice of minimum number of unique residue positions considered. Specifically, we now also report model performance (AUROC and AUPRC) based on a filtered set of ClinVar variants with at least a one-star review status. We have added a new summary table (**Table 1**) and included the following paragraph in the **Results** section:

*“We assessed the robustness of the model in two ways: first, by restricting the analysis to ClinVar variants with at least a one-star review status; and second, by progressively increasing the minimum number of unique residue positions required for EDC calculation. AUROC and AUPRC values under these conditions are summarised in **Table 1**. We found that model performance remained stable when limited to variants with at least a one-star review status. As expected, performance moderately improved when more pathogenic residue positions were considered, reflecting increased confidence in the collective properties of the variants”.*

We additionally note that variants with phenotype annotations are enriched for ‘pathogenic’ labels over ‘likely pathogenic’, with 41% of such variants classified as pathogenic compared to 29% among those lacking phenotype annotations. Therefore the subset of ClinVar variants in our study represents higher-confidence assertions through the consideration of phenotype information.

4) The functional validation is interesting. Why is this not expanded to other DMS assaying stability in ProteinGym?

While ProteinGym and MaveDB include many assays that measure mutation-induced changes in stability, not all stabilising variants have disease relevance or reflect a GOF mechanism in vivo; many

simply improve folding without altering functional activity or driving disease. The *HRAS* and *TP53* assays in the original manuscript represent MAVE assays where DN or GOF variants were confidently detected alongside LOF variants. We have now extended the analysis in **Supplementary Fig. 3** with four additional MAVEs, including *MC4R*, *HMBS*, *PTPN11*, and *MTHFR*, where either GOF variants (in genes known to harbour pathogenic GOF mutations) or hyper-complementing variants (those exhibiting better-than-wild-type fitness in yeast, even if not yet clinically classified as GOF) have been identified. In these assays, using the established classification threshold, we found the mLOF score to consistently discriminate between LOF and non-LOF variants, supporting its use in phenotype-level mechanism prediction.

Additional considerations

1) The authors demonstrate that the mLOF update to the pGOF prior (mLOF + pGOF) results in better performance. In Figure S3D, it is also shown that LoGoFunc outperforms mLOF on its own. Why not assess LoGoFunc for updating the priors (LoGoFunc + pGOF). Does postGOF improve, remain the same, deteriorate?

We initially did not evaluate LoGoFunc-derived scores for updating the pGOF prior due to methodological concerns. LoGoFunc is an ensemble model integrating hundreds of features, many overlapping substantially with those used to construct our gene-level priors (e.g., conservation scores, disorder propensity, homomeric complex formation). Reusing it could double-count evidence, potentially leading to overconfident posteriors. This contrasts with mLOF, which is fully independent of the features used to derive the priors.

However, we have now included this analysis for completeness. The analysis is consistent with the performance of the individual prior and likelihood values: when all genes are considered, pGOF and LoGoFunc_GOF combine to have the highest-performing posterior. Conversely, pGOF and mLOF yield the highest posterior on the test set. We have updated the **Results** section to incorporate this observation:

*“As shown in **Supplementary Figs. 3g-h**, in both cases, when combined with the prior GOF mechanism likelihood, mLOF yielded postGOF scores that substantially outperformed the average GOF probabilities from LoGoFunc. Notably, although updating pGOF with the average GOF probabilities from LoGoFunc achieved the nominally highest AUROC on all data, its performance declined when evaluated on the test set. We also note that LoGoFunc incorporates many features overlapping with those used to derive the gene-level priors, and is therefore not fully independent of the prior, unlike the mLOF score.”*

Reviewer #2 (Remarks to the Author):

Badonyi et al. is building on the findings they described in their previous paper that shows that computational predictors tend to perform poorly on identifying GOF or DN pathogenic variants. Here, they develop a predictive score (mLOF) to distinguish LOF mechanisms from other disease mechanisms. It is an empirical distribution-based approach, which computes the likelihood that a group of missense variants acts via LOF mechanism. The method takes a priori identified pathogenic variants as an input and does not explicitly classify pathogenicity of variants but the mechanism. The mLOF scores are based on their previously described clustering of pathogenic variants (EDC) based on AlphaFold structures and estimated stability effects of mutations using FoldX and computing a normalized mutation effect (ddGrank). The performance of their score is reasonable both on curated clinical data and functional assays of TP53 and HRAS proteins. And it performs better than LoGoFunc method. They share it as a Google Colab notebook that is easy to use, and will be a valuable resource for the biomedical and geneticist community.

Using mLOF scores they find that half of the phenotypes in dominant genes are actually DN or GOF, that is a high estimate.

I find the paper well written, easy to follow with clear figures and representations. The work is novel and interesting and highlights the mechanistic heterogeneity of disease mechanisms even within the same gene posing a challenge for variant interpretation and personalized medicine.

Specific comments:

I would include the definition of gain-of-function and dominant negative effects in the introduction.

We have updated the first paragraph of the **Introduction** to incorporate mechanism definitions:

“Gain-of-function (GOF) mutations cause disease through a wide range of molecular mechanisms, including increased activity (hypermorphs), altered binding specificity, or acquisition of novel functions (neomorphs). Dominant-negative (DN) mutations interfere with the activity of the wild-type protein, either by co-assembling into dysfunctional complexes² or by competitively sequestering shared binding partners or substrates.”

I wonder if a cutoff of 3 variants only is not too low to compute EDC? Please show that the results are robust to the choice of this threshold.

To address this, we have evaluated the robustness of the model to varying this threshold alongside our assessment of ClinVar review status (in response to another reviewer suggestion). We have found that increasing the number of unique residue positions slightly increases performance of the model, likely due to more reliable estimation of the extent of clustering and the energetic impact of the variants. Nonetheless, we believe the baseline cutoff of three positions already provides satisfactory results, and users can expect further gains as more variants become available. We have added a new summary table (**Table 1**) and included the following paragraph in the **Results** section:

*“We assessed the robustness of the model in two ways: first, by progressively increasing the minimum number of unique residue positions required for EDC calculation; and second by restricting the analysis to ClinVar variants with at least a one-star review status. AUROC and AUPRC values under these conditions are summarised in **Table 1**. We found that model performance remained stable when limited to variants with at least a one-star review status. As expected, performance moderately improved when more pathogenic residue positions were considered, reflecting increased confidence in the collective properties of the variants”.*

Regarding performance, please report different metrics at the relevant threshold of 0.508, e.g. sensitivity, specificity, F1 score, accuracy beyond AUROC and precision-recall curve.

We now include the suggested threshold-dependent performance metrics for the mLOF score when the 0.508 threshold is reported to the reader in the **Results** section:

“The resulting value of 0.508 provides a practical cutoff for assessing whether a group of variants is likely to exhibit a LOF mechanism and can be used to compare different variant groups in the same gene. At this threshold, the mLOF score achieves a sensitivity of 0.721, a specificity of 0.702, an accuracy of 0.712, and an F1 measure of 0.719, indicating a balanced performance”.

Line 166: what is a homozygous null variant, please define. This statement does not support the previous claim regarding other mutation types than missense. In general “null variants” and “LOF variants” terms are both used throughout the text. If they are treated as synonyms maybe using only LOF would make the text cleaner.

We thank the reviewer for pointing out the ambiguity. We have clarified in the text that ‘null variants’ refer specifically to stop-gain or frameshift changes that are presumed to abolish protein function entirely:

“Alternatively, missense variants in these phenotypes may follow a single inheritance mode, while other mutation types, such as protein null variants (e.g. nonsense or frameshift mutations that are presumed to completely abolish protein function), particularly when homozygous, correspond to the other mode. This phenomenon has been observed, for example, in ITPR1, where homozygous null and de novo missense variants both cause Gillespie syndrome”.

We avoided using ‘LOF variants’ here to keep mechanistic (LOF) and mutational-consequence (null) concepts distinct, but elsewhere we consistently use ‘LOF’ to refer to the mechanism when contextually appropriate.

Line 185: it is unclear how haploinsufficiency is related to non-LOF mechanisms.

We thank the reviewer for pointing this out and agree that the relationship between haploinsufficiency and non-LOF mechanisms merits clarification. Haploinsufficiency is a specific form of loss-of-function, in which reduced gene dosage (typically from one non-functional allele) is sufficient to cause disease. Once a gene is already associated with a haploinsufficient phenotype, additional disease phenotypes are unlikely to arise through the same dosage-sensitive LOF mechanism. Therefore, additional distinct phenotypes in the same gene are more likely to reflect alternative (non-LOF) mechanisms, such as dominant-negative or gain-of-function effects. We have rewritten the sentence to improve clarity:

“This difference may be explained by the observation that multiple disease phenotypes are unlikely to arise in haploinsufficient genes, where reduced dosage (a form of LOF) already causes disease; thus, by exclusion, additional phenotypes are more likely to involve non-LOF mechanisms”.

The scores for POLG do not seem to be that different or recessive and dominant forms (0.518 and 0.446) and also visually in Figure 3C there is not a big difference in clustering of variants.

We originally chose POLG as an example simply because it showed the highest semantic similarity, but we agree that the difference in clustering between PEOA1 and PEOB1 variants is modest, and that the structural image did not clearly convey the underlying distinction. In this case, the separation in mLOF scores is driven more by differences in predicted stability effects than by spatial clustering. To better illustrate the interpretability of mLOF, we have now replaced POLG with a more compelling example, *CLCN7*, where clustering differences are more visually and quantitatively apparent. Fig.3c and the corresponding paragraph have been updated accordingly:

*“Among these, we highlight *CLCN7* (**Fig. 3c**), with an mLOF score of 0.549 for the recessive and 0.455 for the dominant forms of osteopetrosis (*OPTB4* and *OPTA2*, respectively). These scores are reflective of the considerably greater clustering of dominant variants, with an EDC of 1.36 vs 1.03 for the recessive phenotype. The two phenotypes share many clinical features, as implied by their disease names and semantic similarity scores. Heterozygous osteopetrosis-associated variants are known to exert a dominant-negative effect. The p.Gly215Arg variant, for example, disrupts *CLCN7* trafficking in a dominant-negative manner, and has been used to generate a mouse model of *OPTA2*, which recapitulates the characteristic osteopetrosis phenotype with excessive bone deposition. These findings highlight the utility of combining mLOF scores with semantic similarity to identify DN disease phenotypes in mixed-inheritance genes”.*

It would be interesting to see the distribution of ddG values and not only the ranked (normalized) score as well for the different gene types stratified by inheritance modes (like in Figure 3AB).

We have now added a new analysis in **Supplementary Fig. 4** considering the raw $\Delta\Delta G$ values and including all groups, and a reference to this in the main text:

“We observed that [AR]-AD/AR_{mixed} phenotypes exhibit lower $\Delta\Delta G$ rank values compared with those of

*AR genes ($P = 1.6 \times 10^{-3}$, Wilcoxon rank-sum test), consistent with the presence of hypomorphic variants. A similar tendency was observed using raw $\Delta\Delta G$ values (**Supplementary Fig. 4**). While this trend also appears in AD/AR_{same} genes, we do not have variant-level inheritance classifications in these genes, where the tendency for recessive variants to be hypomorphic may be even stronger, potentially explaining the identical phenotypes for dominant and recessive variants.”*

Since disordered regions (defined by pLDDT <70) and the variants that fall within were excluded from all the analyses, indicating those regions in the structural representation would be helpful, eg. In Figure 4.

We have updated the structures in **Fig. 4** to show pLDDT<70 regions coloured in purple. However, we would like to emphasise that variants falling within these regions are not excluded from the analysis. Instead, these variants are excluded from contributing to the EDC clustering metric, since there could be large variability in their spatial positions. We have added a sentence to the **Discussion** section to make this distinction explicit:

“Second, for the EDC calculation, our method includes variants only within regions with a pLDDT (predicted local distance difference test) score⁸⁹ above 70, limiting it to non-disordered and well-predicted regions of protein structure. Although pathogenic missense mutations are highly enriched in structured regions⁹⁰, this limitation excludes certain variants, e.g., those in short linear motifs⁹¹. Note that variants in disordered regions can still contribute to mLOF via their predicted $\Delta\Delta G$ impacts.”

Please provide the sample size in the caption or plot for Figure S2 and refer to the relevant dataset used. How was balanced precision achieved?

We have updated **Fig. 1** and **Supplementary Fig. 2** to include sample sizes, and added the citation to our earlier study in the caption of **Supplementary Fig. 2**. Balanced precision was calculated as described previously, and we now clarify this in the ‘Statistical analysis’ section: *“Balanced precision was computed by adjusting the standard precision to account for class imbalance, following a previously introduced formula”*.

In Figure S3C and D it is not clear which dataset was used. Please reference it.

In Figure S3A, it is unclear what the mLOF numbers in the legend refer to. Are these the mean values? Maybe showing the distribution would be more meaningful?

In **Supplementary Figs. 3g-h** (previously C-D), each point represents a single-phenotype gene (as defined in this study) annotated with its literature-reported GOF/LOF mechanism label from our previous study, which are now also available in the DECIPHER database. We have added this citation to the figure legend.

In **Supplementary Fig. 3d** (previously A), the mLOF values in the legend represent the missense loss-of-function likelihood scores assigned to variants annotated as LOF, DN or hybrid LOF/DN (referred to as 'mixed LOF/DN' by the original paper) in Giacomelli et al., 2018. The mLOF score is a single value defined for a group of variants, rather than for an individual variant. The important consideration is how these values relate to the established classification threshold.

I do not understand what is shown in Figure S4C and D and I did not find a reference to these panels in the text.

References to **Supplementary Fig. 5c C-D** (previously S4C-D) appear in the 'Phenotype-level analyses' section of the **Methods**. The point of this analysis is to consider that disease phenotypes may have overlapping ClinVar variants, which can be defined by their set relationships. In our analysis of phenotype pairs, we necessarily limited the comparisons to 'distinct' and 'intersect' categories. For full transparency, **Supplementary Fig. 5c** displays the proportions of phenotype pairs in each set-relationship category, both overall as well as stratified by inheritance mode. We have now added more detail to the caption of **Supplementary Fig. 5**:

"As disease phenotypes may share ClinVar missense variants, these can be grouped by their set relationships. This panel shows the distribution of within-gene inheritance pairs based on shared variants, with phenotype pairs classified as distinct, intersecting, subset, or identical. Proportions are shown both overall and stratified by inheritance mode. While our analyses focus on the distinct and intersecting categories, the full distribution is displayed here to provide context. Only inheritance pairs with at least 100 shared variants are included".

I wonder if incorporating structural data on protein complexes would improve the annotation, especially for dominant negative variants that would likely interfere with the assembly of protein complexes for instance. It may be outside of the scope of this work though.

We fully agree that incorporating structural data on protein complexes will be an important direction for future work. In a previous study (Badonyi and Marsh, 2023, *Science Advances*), we used simple regression model to demonstrate that protein complex structural features are valuable for the prediction of non-LOF genes. We have appended the relevant paragraph in the **Discussion** section to bring this to the reader's attention:

"Finally, while we currently rely on monomeric AlphaFold models for EDC and $\Delta\Delta G$ estimation, incorporating predicted structures of protein complexes could substantially improve the accuracy of missense LOF likelihood estimation by providing more biologically representative structural insights through the consideration of intermolecular interactions. For example, our previous work demonstrated that complex properties such as symmetry can be valuable for predicting non-LOF genes, though such features are not yet available for all genes. Improved predictors of protein assembly state and higher-resolution complex structures will likely enhance variant-level predictions.

Furthermore, predicting subunits as part of a complex, rather than in isolation, often yields more accurate conformations due to the presence of buried contacts, potentially making the spatial clustering of pathogenic residues more sensitive and informative".

Reviewer #2 (Remarks to the Author):

The authors have answered all my questions and clarified them in the text. I have no further concerns, only a comment about disordered regions.

Originally, I thought that disordered regions (< pLDDT score 70) were excluded from the analysis. But in the response letter the author stressed that they are only excluded from the EDC analysis.

The authors say (line 438)

“disordered regions can still contribute to mLOF via their predicted $\Delta\Delta G$ impacts”

I am questioning how reliable these $\Delta\Delta G$ values are in these flexible regions that are predicted with low confidence by AlphaFold. Please add some benchmarking or cite previous work that validates the use of FoldX in intrinsically disordered/low pLDDT regions. Or warn the users that these values have to be interpreted with caution.

We have addressed this in the Discussion, at the location the reviewer flagged:

“While these values are difficult to interpret quantitatively due to low confidence in AlphaFold models, mutations in disordered regions almost always receive low $\Delta\Delta G$ values. This is consistent with their limited potential to cause global destabilisation and can still be informative for inferring likely molecular mechanisms.”